

# Gravity waves as a mechanism of troposphere–stratosphere–mesosphere coupling during sudden stratospheric warming

Gordana Jovanovic

University of Montenegro, Faculty of Science and Mathematics, Montenegro

Dzordza Vasingtona bb, 81000 Podgorica, Montenegro

**Correspondence:** Gordana Jovanovic  (gordanaj@ucg.ac.me)

**Abstract.** The propagation of gravity waves (GW) and their role in the coupling of the troposphere–stratosphere–mesosphere atmospheric layers during sudden stratospheric warming (SSW) are studied. A standard set of hydrodynamic equations (HD) is used to derive the analytical dispersion equations and the GWs reflection coefficient. These equations are applied to the troposphere–stratosphere and stratosphere–mesosphere boundaries to analyze which part of the GWs spectra has the greatest

chance of crossing them and affecting the dynamics of the upper atmosphere. We found that the GW reflection coefficient at the troposphere–stratosphere boundary increases significantly during SSW. This is not the case for the reflection coefficient at the stratosphere–mesosphere boundary when the reflection coefficient decreases compared to its value in the no–SSW case. The generation of GWs in the stratosphere during the SSW is responsible for the reduction of the reflection coefficient. However, these additional GW fluxes are not sufficient to compensate for the reduction of GW fluxes coming from the troposphere to the

mesosphere. As a result, there is mesospheric cooling accompanied by SSW events.

## 1 Introduction

The stratosphere is a part of the Earth's atmosphere, embedded between the troposphere and the mesosphere at an altitude of about 10 to 50 km. It is a stably stratified medium which enables the propagation of the acoustic–gravity waves. Its temperature varies from $\sim 220$ K at the lower boundary to $\sim 270$ K at the upper boundary. Temperature increases with ozone concentration,

as solar energy is converted to kinetic energy when ozone molecules absorb ultraviolet radiation (UV), resulting in heating of the stratosphere. Ozone is formed naturally and photochemically within the stratosphere (the Chapman cycle). It is considered a pollutant in the troposphere, but in the stratosphere it is essential to life on Earth because it absorbs biologically harmful UV radiation. The warming of the stratosphere can occur through another mechanism known as sudden stratospheric warming (SSW). It is a rapid warming when the temperature rises by about 50 K in just a few days. SSWs are caused by the breaking

of planetary–scale (Rossby) waves and gravity waves that propagate upwards from the troposphere (Cullens and Thurairajah , 2021). During an SSW, the polar vortex breaks down, accompanied by rapid descent and warming of air in polar latitudes, mirrored by ascent and cooling above the warming. The rapid warming and descent of the polar air affect tropospheric weather, shifting jet streams, storm tracks, and the Northern Annular Mode, making cold air outbreaks over North America and Eurasia





more likely (Zhang and Chen, 2019). This phenomenon mainly occurs in winter and spring, about six times per decade. SSW events can be devided into major and minor events based on their warming intensity, according to whether an event causes the polar circulation to reverse. Warmings are commonly classified as "minor" when the zonal–mean 10–hPa meridional temperature gradient between $60^0$ N and $90^0$ N reverses, and as "major" when in addition the zonal–mean 10–hPa zonal wind at $60^0$ N reverses (Stephan et al. , 2020; Gogoi et al. , 2023). SSWs affect the atmosphere above and below the stratosphere, producing widespread effects on atmospheric chemistry, temperatures, winds, neutral (nonionized) particles and electron densities (Matsuno , 1971; Baldwin et al. , 2021; Rupp et al. , 2023). Therefore, SSWs are the most prominent manifestation of connections between the lower, middle, and upper atmosphere and a proper and detailed study of such events is important for understanding the interactions between different atmospheric layers (Goncharenko et al. , 2012; Gupta and Upadhayaya, 2017; Goncharenko et al. , 2018; Domeisen, 2019). These effects span both hemispheres (de Jesus et al. , 2017; Zhang and Chen, 2019; Wang et al. , 2020; Liu et al. , 2022; Mariaccia, Keckhut, and Hauchecorne , 2022). The stratospheric changes during SSWs modulate the spectrum of atmospheric waves that propagate through the stratosphere and upward into the mesosphere. In this article, the focus is on atmospheric gravity waves, which are part of the acoustic–gravity waves spectra. Namely, it is known that acoustic waves, unlike GWs, are strongly absorbed in the atmosphere (Sindelarova, Buresova, and Chum , 2009). The rate of absorption is proportional to the wave frequency squared. Therefore, only GWs are in the focus of this article. Gravity waves (GWs) exist over a wide range of horizontal scales and typically have time scales short enough to ignore rotation, heat transfer and friction (Köhler , 2020). They are usually categorised by their source of origin, which can be orography (Minamihara et al. , 2016) or synoptic systems such as convection (Vincent and Alexander , 2000), jets or fronts (Fritts and Alexander , 2003; Plougonven and Zhang , 2014). These waves typically propagate from the troposphere through the stratosphere into the mesosphere. With exponential amplitude growth, the gravity waves will have grown so large that they become unstable and break, thereby altering the atmospheric flow by depositing stored momentum and energy (Kalisch and Chun , 2021). SSW is connected with a strong mesospheric cooling because a filtering of gravity waves during SSW events induces strong mesospheric cooling (Holton , 1983). In their study, Cullens and Thurairajah (2021) analyzed 40–years of long–term ERA5 output in order to study the general trends in GWs variations before, during, and after the SSW.

In this article, the impact of stratospheric temperature change on GWs characteristics is studied. We analyzed the upward propagation of GWs through the Earth's atmosphere, modeled by two different temperature layers separated by a horizontal plane boundary. Analytical equation for the reflection coefficient is derived and applied to the troposphere–stratosphere and stratosphere–mesosphere boundaries under the normal atmospheric conditions and during SSW event. Two important points can be distinguished: the first is that GWs coming from the troposphere into the stratosphere participate in the generation of SSWs, and the second is that GWs generated in the stratosphere during SSWs also participate in the mesospheric dynamics.

## 2 Basic equations

The standard set of hydrodynamic equations (HD) describes the dynamics of adiabatic processes in the atmosphere stratified by the presence of gravity with constant acceleration $g = 9.81 m/s^2$:





continue and ideal gas equation

$$\frac{\partial \rho}{\partial t} + \nabla \cdot (\rho \boldsymbol{v}) = 0, \quad p = \rho R T, \tag{1}$$

momentum equation

$$\rho \left( \frac{\partial \boldsymbol{v}}{\partial t} + \boldsymbol{v} \cdot \nabla \boldsymbol{v} \right) = -\nabla p + \rho \boldsymbol{g} \tag{2}$$

and an adiabatic law for a perfect gas

$$\frac{\partial p}{\partial t} + \boldsymbol{v} \cdot \nabla p = \frac{\gamma p}{\rho} \left( \frac{\partial \rho}{\partial t} + \boldsymbol{v} \cdot \nabla \rho \right). \tag{3}$$

Here, $R = R_0/M$ is the individual gas constant for molecules with molar mass M, $R_0 = 8.314 J/mol K$ is the universal gas constant and $\gamma = c_p/c_v = (j+2)/j$ is the ratio of specific heats for gas particle with $j = 5$ degrees of freedom.

## 2.1 Dispersion equation for acoustic–gravity waves (AGWs)

The dispersion equation relates the wave frequency to the wave numbers (wave's spatial characteristics) and to the background atmosphere properties. We consider waves whose wavelengths are sufficiently small in comparison with the Earth radius $R_E =$
6371 km. Therefore, the plane parallel geometry can be applied in a locally isothermal medium. Under these assumptions, the atmosphere is taken to be vertically stratified, initially in hydrostatic equilibrium, and then perturbed by harmonic waves of small amplitude. This means that Eqs. (1)–(3) can be linearized by taking any physical quantity $\psi(x,y,z,t)$ as a sum of its basic state unperturbed value $\psi_0(z)$ and a small first order perturbation $\delta\psi(x,y,z,t)$, i.e. $\psi(x,y,z,t) = \psi_0(z) + \delta\psi(x,y,z,t)$, where: $\delta\psi(x,y,z,t) = \psi^{'}(z)e^{i(k_x x + k_y y - \omega t)}$, and $|\psi^{'}| \ll |\psi_0|$. Eqs. (1)–(3), linearized with these perturbations, reduce to three
equations: one for the unperturbed basic state and two for small perturbations. The unperturbed basic state is desribed by:

$$\frac{d}{dz} \ln \rho_0(z) + \frac{1}{H} = 0, \quad p_0 = \rho_0 R T_0, \quad \text{with} \quad T_0 = const,$$

whose solution is:

$$\rho_0(z) = \rho_0(0)e^{-z/H} \quad \text{or} \quad p_0(z) = p_0(0)e^{-z/H}, \tag{4}$$

where $H = p_0(0)/\rho_0(0) = v_s^2/\gamma g = const$ is the characteristic scale–height of the isothermal atmosphere.
The small perturbations are governed by equations (Jovanovic , 2016):

$$\frac{d\xi_z^{'}}{dz} = C_1 \xi_z^{'} - C_2 p^{'}, \quad \frac{dp^{'}}{dz} - g\frac{d\rho_0}{dz}\xi_z^{'} = C_3 \xi_z^{'} - C_1 p^{'}, \tag{5}$$

where $\xi_z^{'} = iv_z^{'}/\omega$ is the z–component (i.e. the vertical component) of the fluid displacement, while $p^{'}$ is the pressure perturbation. The coefficients in Eqs. (5) are:

$$C_1 = \frac{g}{v_s^2}, \quad C_2 = \frac{\omega^2 - k_p^2 v_s^2}{\rho_0(z) v_s^2 \omega^2}, \quad C_3 = \rho_0(z) \left( \omega^2 + \frac{g^2}{v_s^2} \right). \tag{6}$$



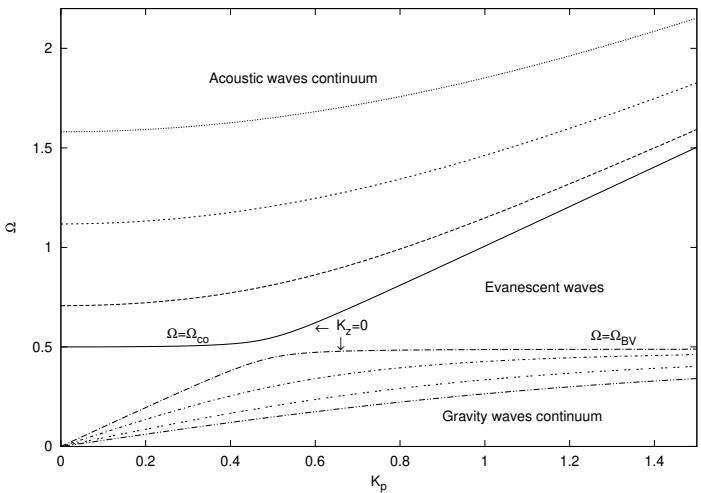

**Figure 1.** Dispersion curves for AGWs. Two sets of curves are related to acoustic and gravity waves, which cannot propagate below the acoustic cutoff frequency $\Omega_{co} = \omega_{co}H/v_s$ and above the Brunt–Väisälää frequency $\Omega_{BV} = \omega_{BV}H/v_s$, respectively.

The density distribution $\rho_0(z)$ is given by Eq. (4) and $k_p^2 = k_x^2 + k_y^2$ designates square of the horizontal wavenumber. The Eqs. (5)–(6) allow the following solutions for the vertical displacement $\xi_z^{'}$ and the pressure perturbation $p^{'}$:

$$\xi_z^{'}(z) = \xi_z^{'}(0)e^{\frac{z}{2H}}e^{ik_z z}, \ \ p^{'}(z) = p^{'}(0)e^{\frac{-z}{2H}}e^{ik_z z}. \tag{7}$$

Eqs. (5) with solutions Eqs. (7) yield the dispersion equation for AGWs:

$$k_z^2 = \frac{\omega^2(\omega^2 - \omega_{co}^2) - k_p^2 v_s^2(\omega^2 - \omega_{BV}^2)}{\omega^2 v_s^2}. \tag{8}$$

Here, $k_z$ is the vertical wavenumber, $\omega_{co}^2 = \gamma^2 g^2/4v_s^2 = v_s^2/4H^2$ is the square of the acoustic wave cutoff frequency, and $\omega_{BV}^2 = (\gamma - 1)g^2/v_s^2$ is the square of the Brunt–Väisälää frequency. This equation is quadratic in $\omega^2$ which indicates the existence of two wave modes in the considered stratified atmosphere: the acoustic and gravity modes. Stratification in a vertical direction, caused by gravity and given by Eq. (4), introduces cutoff frequencies–acoustic cutoff frequency below which acoustic waves cannot propagate and Brunt–Väisälää frequency above which gravity waves cannot propagate. Therefore, the branches

of acoustic and gravity waves are present. Between them are evanescent waves that do not propagate, Fig. 1. The physical quantities in the dispersion equation can be made dimensionless by appropriate scalings: $K_p = k_p H$, $K_z = k_z H$, $\Omega = \omega H/v_s$, $\Omega_{co} = \omega_{co}H/v_s = 0.5$ and $\Omega_{BV} = \omega_{BV}H/v_s = \sqrt{\gamma - 1}/\gamma = 0.45$. Now, the dispersion equation Eq. (8) has the dimensionless form:

$$K_z^2 = \Omega^2 - \Omega_{co}^2 - \frac{K_p^2(\Omega^2 - \Omega_{BV}^2)}{\Omega^2}. \tag{9}$$

The AGWs propagate in the vertical direction if $K_z^2 > 0$. This is fulfilled when

$$K_p^2 < \frac{\Omega^2(\Omega^2 - \Omega_{co}^2)}{\Omega^2 - \Omega_{BV}^2}, \tag{10}$$





i.e. when dimensionless horizontal phase velocity is

$$V_h^2 = \frac{\Omega^2}{K_p^2} > \frac{\Omega^2 - \Omega_{BV}^2}{\Omega^2 - \Omega_{co}^2}. \tag{11}$$

The AGWs become evanescent when $K_p^2 > \frac{\Omega^2(\Omega^2 - \Omega_{co}^2)}{\Omega^2 - \Omega_{BV}^2}$ and $V_h^2 < \frac{\Omega^2 - \Omega_{BV}^2}{\Omega^2 - \Omega_{co}^2}$. The boundary between propagating and evanes-
cent regions is defined by $K_z = 0$. Gravity waves, in contrast to acoustic waves, are not able to travel vertically with $K_p = 0$.
This means there are no pure vertically propagating gravity waves (Mihalas , 1984). Therefore, they propagate obliquely
through the stratified atmosphere in accordance with the dispersion equation. Dimensionless equations are used because they
are valid in each stratified medium, like Earth's, planet's or the solar atmosphere. When we rewrite them using characteristic
frequencies and temperatures, we obtain the equations for particular atmospheric layers.

## 3 Reflection coefficient of GWs

The considered basic state in the stratified atmosphere is composed of two half–spaces with constant sound speeds, separated
by a horizontal plane boundary $z = 0$. The two regions are characterized by the corresponding plasma densities $\rho_{01}$ and $\rho_{02}$
adjacent to the lower and upper side of the boundary $z = 0$. The unperturbed density profile can be expressed as follows:

$$\rho_0(z) = \rho_{01} e^{-z/H_1}, \ z < 0, \ \text{region (1)}, \quad \rho_0(z) = \rho_{02} e^{-z/H_2}, \ z > 0, \ \text{region (2)}, \tag{12}$$

where $H(n) = v_{sn}^2/\gamma g$, $n = 1, 2$. There is a density, pressure, and temperature jump across $z = 0$. The boundary condition that
has to be applied at $z = 0$ in the basic state is the continuity of the unperturbed pressure $p_0$ at $z = 0$, (Jovanovic , 2016), which
yield:

$$\frac{\rho_{02}}{\rho_{01}} = \frac{v_{s1}^2}{v_{s2}^2} = \frac{T_1}{T_2} = s = const. \tag{13}$$

The boundary conditions for perturbations are continuity of both the vertical fluid displacement $\xi_z'$ and the pressure perturbation
$p' - g\rho_0(z)\xi_z'$ at the boundary $z = 0$. Also, the energy density of the perturbations has to diminish to zero as $|z|$ tends to infinity.
The harmonic wave, which propagates through regions (1) and (2), does not change its frequency, and the horizontal wavevector
component $K_p$, parallel to the boundary $z = 0$. However, the vertical wavevector component $K_z$ has a discontinuity at the
boundary $z = 0$, where it changes from $K_{z1}$ to $K_{z2}$ according to the dispersion equation Eq. (9). We assume that a wave
propagates from the lower region (1) upward towards the boundary $z = 0$, and that the waves continuing past it are absorbed
with no reflection in the upper region (2). In this case, in the lower region, the perturbations are the superposition of the incident
and reflected waves, while in the upper region, there is only the transmitted wave. The reflection coefficient of AGWs is defined
as the square of the absolute value of the reflection amplitude. Using dimensionless physical values for brevity, the reflection
coefficient can be written as (see details in Jovanović , 2014):

$$R = \left[ \frac{\left[ \left(1 - \frac{\gamma}{2}\right)\left(\frac{1}{V_h^2 - 1} - \frac{s^2}{sV_h^2 - 1}\right) + \frac{(s-1)}{V_h^2}\right]^2 + \frac{\gamma^2\Omega^2}{V_{v1}^2}\left(\frac{V_{v1}^2}{V_{v2}^2} \cdot \frac{s^2}{(sV_h^2 - 1)^2} - \frac{1}{(V_h^2 - 1)^2}\right)}{\left[\left(1 - \frac{\gamma}{2}\right)\left(\frac{1}{V_h^2 - 1} - \frac{s^2}{sV_h^2 - 1}\right) + \frac{(s-1)}{V_h^2}\right]^2 + \frac{\gamma^2\Omega^2}{V_{v1}^2}\left[\frac{V_{v1}}{V_{v2}} \cdot \frac{s}{sV_h^2 - 1} + \frac{1}{V_h^2 - 1}\right]^2} \right]^2 +$$




$$\left[\frac{\frac{2\gamma\Omega}{V_{v1}(V_h^2-1)}\left[\left(1-\frac{\gamma}{2}\right)\left(\frac{1}{V_h^2-1}-\frac{s^2}{sV_h^2-1}\right)+\frac{(s-1)}{V_h^2}\right]}{\left[\left(1-\frac{\gamma}{2}\right)\left(\frac{1}{V_h^2-1}-\frac{s^2}{sV_h^2-1}\right)+\frac{(s-1)}{V_h^2}\right]^2+\frac{\gamma^2\Omega^2}{V_{v1}^2}\left[\frac{V_{v1}}{V_{v2}}\cdot\frac{s}{sV_h^2-1}+\frac{1}{V_h^2-1}\right]^2}\right]^2. \tag{14}$$

Here, $V_{v1}$ and $V_{v2}$ are the vertical phase velocities of AGWs in regions (1) and (2) respectively, given by the equations:

$$V_{v1}=\frac{\Omega}{K_{z1}}=\frac{V_h\Omega}{\sqrt{V_h^2(\Omega^2-\Omega_{co}^2)-(\Omega^2-\Omega_{BV}^2)}}, \tag{15}$$

and

$$V_{v2}=\frac{\Omega}{K_{z2}}=\frac{V_h\Omega}{\sqrt{sV_h^2(\Omega^2-s\Omega_{co}^2)-(\Omega^2-s\Omega_{BV}^2)}}, \tag{16}$$

while $V_h$ is horizontal phase velocity given by Eq. (11). If $V_{v1}^2$ and $V_{v2}^2$ are positive, AGWs propagate through regions (1) and (2), respectively. If $V_{v1}^2, V_{v2}^2 < 0$, these waves are evanescent and not of interest to this study.

## 4 Results

The analytical equations derived in Sect. 2 and Sect. 3 are used to analyze the propagation of GWs and their reflection/-
transmission properties at the troposphere–stratosphere and stratosphere–mesosphere boundaries. Gravity waves can reach the stratosphere from below, but they can also be excited in the stratosphere during a minor SSW (Dörnbrack et al. , 2018). This source mechanism to generate GWs is known as spontaneous adjustment (Plougonven and Zhang , 2014). Excited in situ within the stratosphere, GWs can propagate upward toward the mesosphere.

In the stratosphere with a temperature $T = 240K$ and $\gamma = 1.4$, sound velocity is $v_s = \sqrt{\gamma RT} = 310m/s$ and scale–height is
$H = 7000m$. The Brunt–Väisälää frequency is $\omega_{BV} = \sqrt{\gamma - 1}g/v_s = 0.02s^{-1}$. During SSW, the temperature in the strato-sphere can rise by more than 50 K, i.e. $T = 290K$. Sound velocity is now $v_s = 341m/s$, scale–height is $H = 8467m$ and the Brunt–Väisälää frequency is lower than before SSW, i.e. $\omega_{BV} = 0.018s^{-1}$.

### 4.1 Gravity waves at the troposphere–stratosphere boundary

Gravity waves can propagate through both regions–the troposphere and the stratosphere if $V_{v1}^2$ and $V_{v2}^2$ in Eqs.(15) and (16) are
positive, i.e. if $\Omega < \sqrt{s}\Omega_{BV} = 0.43$, or $\omega < 0.02s^{-1}$ and $V_h < \Omega_{BV}/\Omega_{co} = 0.9$, or $v_h < 267m/s$. The reflection coefficient for GWs which propagate from the troposphere whose temperature is $T_1 \approx 220K$ (this is the temperature at the tropopause) towards the stratosphere whose temperature is $T_2 \approx 240K$ is presented in Fig. 2. Here the parameter $s = T_1/T_2$ has the value $s = 0.91$. The reflection coefficient increases with increasing frequency $\Omega$ and with decreasing horizontal phase velocity $V_h$. Its value remains below 0.4 for GWs with very low frequencies, $\Omega < 0.2$, i.e. $\omega < 0.009s^{-1}$ and with $0.1 < V_h < 0.9$, i.e.
$30m/s < v_h < 267m/s$, Fig. 2, area within the rectangle. When SSW starts, the stratospheric temperature can rise from 240 K to 290 K within a few days. Now the parameter s is $s = T_1/T_2 = 0.76$. Due to the temperature change during SSW, the





frequency range for propagating GWs also change. It is reduced from the $\Omega < \sqrt{0.91}\Omega_{BV} = 0.43$, i.e. $\omega < 0.02s^{-1}$ to $\Omega < \sqrt{0.76}\Omega_{BV} = 0.39$, i.e. $\omega < 0.018s^{-1}$. Temperature change also affects the reflection coefficient of GWs, Fig. 3. An increase in the reflection coefficient of gravity waves propagating from the troposphere to the stratosphere during the SSW is obvious.

GWs with $\Omega < 0.1$, i.e. $\omega < 0.005s^{-1}$ and $0.3 < V_h < 0.9$, i.e. $90m/s < v_h < 267m/s$ have the reflection coefficients below 0.4 and the best chance to pass troposphere–stratosphere boundary, Fig. 3, area within the rectangle. It is evident that the frequency band for GWs to cross the troposphere-stratosphere boundary is decreased.

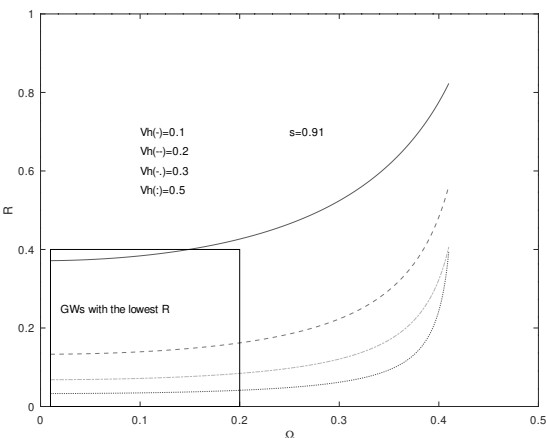

**Figure 2.** The reflection coefficient of gravity waves propagating from the troposphere to the stratosphere under normal stratospheric conditions as a function of frequency, with the horizontal phase velocity and $s = T_1/T_2 = 0.91$ as parameters. The rectangular region shows the range of frequencies and horizontal phase velocities of GWs with $R < 0.4$.

**Figure 3.** The reflection coefficient of gravity waves propagating from the troposphere to the stratosphere during SSW as a function of frequency, with the horizontal phase velocity and $s = T_1/T_2 = 0.76$ as parameters. The rectangular region shows the range of frequencies and horizontal phase velocities of GWs with $R < 0.4$.

## 4.2  Gravity waves at the stratosphere–mesosphere boundary

Gravity waves originating in the stratosphere can travel to the mesosphere. Under normal conditions, the temperature in the
stratosphere is $T_1 = 240K$, while the temperature at the stratosphere–mesosphere boundary is $T_2 = 270K$ (this is the temperature at the stratopause). Thus, parameter $s = T_1/T_2 = 0.89$. GWs can propagate in both regions–the stratosphere and the mesosphere if $\Omega < \sqrt{s}\Omega_{BV} = 0.42$, i.e. $\omega < 0.019s^{-1}$ and $V_h < 0.9$, i.e. $v_h < 279m/s$. The dimensionless horizontal phase velocity has the same value $V_h < 0.9$ as in the case when GWs propagate from the troposphere towards the stratosphere. Knowing that $V_h = \Omega/K_p = v_h/v_s$, it is obvious that the horizontal phase velocity $v_h$ depends on the sound velocity $v_s$ in
a specific atmospheric layer. Therefore, GWs that propagate through the troposphere–stratosphere boundary have a horizontal phase velocity $v_h < 267m/s$, while GWs that propagate through the stratosphere–mesosphere boundary have a horizontal



phase velocity $v_h < 279m/s$.

The reflection coefficient for GWs at the stratosphere–mesosphere boundary in normal stratospheric conditions is presented in Fig. 4. As in Fig. 2, it increases with increasing frequency $\Omega$ and with decreasing horizontal phase velocity $V_h$. GWs with $\Omega < 0.2$, i.e. $\omega < 0.009s^{-1}$ and $0.1 < V_h < 0.9$, i.e. $31m/s < v_h < 279m/s$, whose reflection coefficients are below 0.4, are the best candidates to enter the mesosphere, Fig. 4, rectangular region.

During the SSW, the stratospheric temperature rises to 290 K, causing a change in the parameter $s = T_1/T_2$, which becomes $s = 1.1$. This changes the conditions for GWs propagation. They propagate in both regions–the stratosphere and the mesosphere if $\Omega < \sqrt{s}\Omega_{BV} = 0.45$, i.e. $\omega < 0.018s^{-1}$ and $V_h = \Omega_{BV}/\sqrt{s}\Omega_{co} < 0.86$, i.e. $v_h < 293m/s$. The reflection coefficient of GWs in this case is shown in Fig. 5. Comparing Figs. 4 and 5, it can be seen that the reflection coefficient decreases durinig SSW and therefore GWs can cross the stratosphere–mesosphere boundary more easily than under normal stratospheric conditions. This especially refers to GWs with frequencies $\Omega < 0.2$ or $\omega < 0.008s^{-1}$, and with horizontal phase velocities $0.1 < V_h < 0.86$, i.e. $34m/s < v_h < 293m/s$, whose $R < 0.4$, Fig. 5, rectangular region. Note that the dimensionless frequency has the same value $\Omega < 0.2$ as in the case when GWs propagate from the troposphere to the stratosphere in the no–SSW situation. Knowing that $\Omega = \omega H/v_s$, it is obvious that the frequency $\omega$ depends on the sound velocity $v_s$ and characteristic scale–height H in a specific atmospheric layer. Therefore, GWs that propagate through the troposphere–stratosphere boundary have a frequency $\omega < 0.009s^{-1}$, while GWs that propagate through the stratosphere–mesosphere boundary have a frequency $\omega < 0.008s^{-1}$. The situation is similar for GWs that propagate through the stratosphere–mesosphere boundary in normal stratospheric conditions when $\Omega < 0.2$ means $\omega < 0.009s^{-1}$, and during SSW events when $\Omega < 0.2$ means $\omega < 0.008s^{-1}$.

## 5   Discussion

SSWs trigger a chain of events that lead to anomalies in the stratosphere, and thus to anomalies in the adjacent layers–the troposphere and mesosphere. Stratospheric anomalies are caused mainly by wave forcing from the dense troposphere. Two types of waves that play an important role for the stratospheric variability are gravity waves and planetary (Rossby) waves. GWs considered in this article exist in a stably stratified atmosphere. Their characteristics and reflection/transmission properties in the Earth's and solar atmosfere are described in the scientific literature (Marmolino et al. , 1993; Jovanovic , 2016; Fleck et al. , 2020). GWs that propagate from the troposphere to the stratosphere affect the generation of SSW. The reflection coefficient shown in Fig. 2 indicates that GWs with small frequencies $\omega < 0.009s^{-1}$, about 2 times smaller than the Brunt–Väisälää frequency $\omega_{BV} = 0.02s^{-1}$, can penetrate the stratosphere and influence its dynamics. Albers and Birner  (2014) found that these GWs can contribute to the occurrences of SSWs up to $30\%$. This result is confirmed in the works of Cullens and Thurairajah  (2021) and Gupta et al.  (2021). During SSWs, the temperature in the stratosphere increases by about 50 K. Fig. 3 shows that SSW events prevent GWs propagation from the troposphere towards the stratosphere, which is consistent with known scientific results (Wang and Alexander , 2009; Hindley et al. , 2020; Wicker, Polichtchouk, and Domeisen , 2023). GWs with the reflection coefficient $R < 0.4$ have frequencies $\omega < 0.005s^{-1}$ and horizontal phase velocities $90m/s < v_h < 267m/s$. These waves are the best candidates for passing through the troposphere–stratosphere boundary. Note that the frequency range for




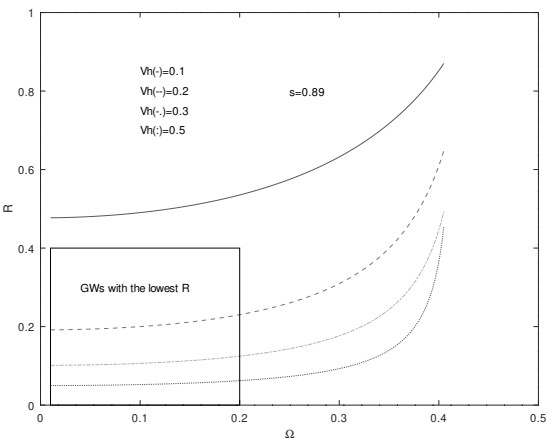

**Figure 4.** The reflection coefficient of gravity waves propagating from the stratosphere to the mesosphere under normal stratospheric conditions as a function of frequency, with the horizontal phase velocity and $s = T_1/T_2 = 0.89$ as parameters. The rectangular region shows the range of frequencies and horizontal phase velocities of GWs with $R < 0.4$.

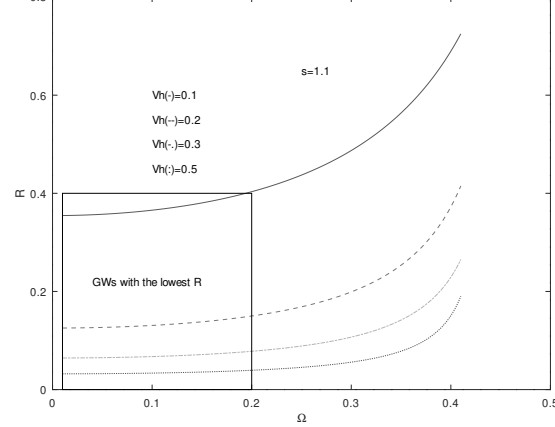

**Figure 5.** The reflection coefficient of gravity waves propagating from the stratosphere to the mesosphere during SSW as a function of frequency, with the horizontal phase velocity and $s = T_1/T_2 = 1.1$ as parameters. The rectangular region shows the range of frequencies and horizontal phase velocities of GWs with $R < 0.4$.

GWs transmission is reduced from $\omega < 0.009 s^{-1}$ in the no–SSW case, to $\omega < 0.005 s^{-1}$ in the SSW case. This means that the frequency band for GWs transmission from the troposphere to the stratosphere is narrower.

Disruption of the polar vortex during the SSW events allows cold air to descend from the stratosphere to the troposphere and move it from the pole to the mid–latitudes. These changes affect the climate and may lead to a dramatic decrease in temperature in Northern Europe (Baldwin et al. , 2001; King et al. , 2019). This confirms the existence of the two–way stratospheric–tropospheric dynamical coupling (Mariaccia, Keckhut, and Hauchecorne , 2022). In addition, SSW–induced temperature changes can modify chemical reaction rates, which is particularly important for upper stratospheric ozone (Pedatella et al. , 2018).

The inhibition of GWs propagating upward from the troposphere to the stratosphere, Fig. 3, and the causal absence of gravity wave breaking in the mesosphere explains the mesospheric cooling during an SSW (Holton , 1983; Liu and Roble , 2002). Also, the mesospheric wind changes are related to the ways that the stratosphere influences the filtering of GWs (Pedatella et al. , 2018: Kalisch and Chun , 2021). Therefore, the state of the stratosphere is important for the propagation of GWs in the upper atmosphere. It varies when the SSW starts. While an increase in the reflection coefficient at the troposphere–stratosphere boundary was expected, Figs. 2 and 3, the decrease in the reflection coefficient at the stratosphere–mesosphere boundary requires an explanation. We think that the generation of GWs in the stratosphere, in situ, during SSW increases the possibility of these waves penetrating the mesosphere. This could be the reason for the lower reflection coefficient compared to the case without SSW. Although GWs generated in the stratosphere contribute to mesospheric dynamics and temperatures, they cannot





compensate for the strong reflection of GWs generated in the troposphere at the troposphere-stratosphere boundary, Fig. 3. The result is a detected mesospheric cooling. This cooling is strongest for the GWs with $\Omega > 0.2$ or $\omega > 0.008 s^{-1}$ and with $V_h < 0.1$ or $v_h < 34 m/s$ because these waves have the least chance of crossing the stratosphere-mesosphere boundary and

penetrating the mesosphere, Fig. 5. Their reflection coefficient is $R > 0.4$. This is in agreement with the strongest mesospheric cooling found in Stephan et al. (2020).

Changes in the stratosphere are also caused by solar activity. Namely, in the Earth's atmosphere, the solar spectral irradiance (SSI) forcing plays a key role as the main driver in the so called top–down mechanism (Gray et al. , 2010; Tsuda, Shepherd, and Gopalswamy , 2015). This mechanism originates in the stratosphere, where UV radiation modulates local radiative heating

at the tropical stratopause and ozone chemistry. In addition, the SSI directly impacts the ultra–violet (UV) photolysis of $O_2$, an important source of ozone in the stratosphere. The potential drop/rise in the solar UV activity can substantially affect the ozone layer, which in turn affects stratospheric temperature, circulation, tropospheric climate, and the UV intensity reaching the ground (Anet et al. , 2013). In the upper stratosphere satellite observations show an increase in temperature of 1–2 K from the solar minimum to solar maximum activity during the 11-year solar cycle (Ineson et al. , 2011). Note that this temperature

increase is much smaller than the temperature increase of about 50 K during SSW that occurs in a few days. Therefore, the analysis presented in this article is not applicable to stratospheric temperature changes during the 11-year solar cycle.

Disturbances in the stratosphere and changes in GW propagation during SSW events affect the electron concentration in the lower ionosphere. Namely, in the presence of GWs the electron concentration becomes time–dependent and this influences the reflection of very low frequency waves (VLF), as studied in Nina and Čadež (2013), and Nina et al. (2017) with the

consequences in telecommunications and navigation. It appears that the SSWs can be considered within the framework of the atmosphere–ionosphere system (Yiğit and Medvedev , 2016).

## 6   Conclusions

SSWs have a long-lasting effect within the stratosphere, as well as an impact on the adjacent troposphere and mesosphere. SSWs impact the tropospheric circulation confirming the existence of the stratospheric–tropospheric dynamical coupling

(Mariaccia, Keckhut, and Hauchecorne , 2022). During the SSW events, the reflection coefficient for GWs at the troposphere–stratosphere boundary increases significantly, Fig. 3. This filtration of GWs has a major impact on mesospheric dynamics because generation of GWs in the stratosphere during SSW cannot compensate the reduction in the GWs from the troposphere. Therefore, during SSW we have the two accompanied processes–stratospheric warming and mesospheric cooling. GWs are the coupling mechanism between these two processes. We used HD equations and temperature as the main

parameter to derive the dispersion equation for GWs and their reflection coefficient. An increase in the reflection coefficient at the troposphere–stratosphere boundary, i.e. an increase in downward GW fluxes can be used to predict SSW events similar as in Rupp et al. (2023). Detailed knowledge of how stratospheric anomalies influence tropospheric weather will open the door to improved climate models and forecasts. The effects of SSWs on the upper atmosphere will enable scientists to improve space weather forecasting, especially to determine day–to–day variability in the ionosphere (Yiğit and Medvedev , 2016). The phys-



ical processes that contribute to the variability of the Earth's atmospheric layers also operate in other planetary atmospheres and define their dynamics and energy budgets. Therefore, information gained from this study of the coupling between Earth's atmospheric layers is potentially applicable to atmospheres of other planets.

*Author contributions.* There is only one author and the article is the result of the work of one author.

*Competing interests.* The author declares that they have no conflict of interest.

*Acknowledgements.* The research and writing of this work was supported by the Montenegrin National Project "Physics of Ionized Gases and Ionized Radiation".



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
