# Peer review of "Gravity waves as a mechanism of troposphere–stratosphere–mesosphere coupling during sudden stratospheric warming"

_EGUsphere, 2024_

## Referee Comment (RC1)

This paper studies the coupling between the troposphere, stratosphere, and mesosphere by gravity waves during a sudden stratospheric warming (SSW). By using an analytical reflection coefficient R to characterise the spectrum of waves that is transmitted/reflected at the tropopause and stratopause during SSW events, the study offers a novel perspective on the role of gravity waves in atmospheric coupling and the observed effect of a cooling mesosphere during an SSW. Given the knowledge gaps in this area and the need for better representation of gravity wave effects in models, I believe this paper adds value and recommend it for publication provided that the following comments are addressed.

- Whilst the data in the figures supports the conclusion drawn regarding the changes in R during this idealised SSW, I am confused by the key facts used to illustrate the takeaway points. Firstly, please could you provide a reason for the R<0.4 threshold used to delineate transmitted/reflected waves in figures $2 - 5$, it seems like an arbitrary choice. Please then could you provide a reason for the $\Omega = 0.2$ and $\Omega = 0.1$ thresholds in Figs 2/4/5 and Fig 3, respectively? $\Omega < 0.2$ and $\Omega < 0.1$ mark most likely transmitted waves given $R = 0.4$, however the $\Omega$ thresholds don't appear to reflect what the figure tells you. For example, in Fig. 2, almost all $\Omega$ have $R < 0.4$ for $V_h < 0.5$, the text in line 149 however implies that only $\Omega < 0.2$ have $R < 0.4$ for $V_h < 0.9$. The same critique applies to Fig 3 and L 155, Fig 4 and L 170, Fig 5 and L 177. These numbers are used in the discussion in the paper, so are a significant aspect which needs addressing before the paper is published.

- The assumptions used in deriving the reflection coefficient, and the set-up of the two scenarios: no-SSW, and SSW, are highly idealised. I.e., The atmosphere from the tropopause to the mesosphere is not isothermal and is highly variable compared to the two idealised scenarios given. Please could you address the sensitivity of your results to deviations from the assumptions and idealisations, and the validity of using this form of the reflection coefficient to the real atmosphere? Perhaps you could support its validity by calculating R from data of observed or modelled SSW events, which should hopefully show that R increases at the tropopause, decreases at the stratopause, and in such a way that suggest similar changes in gravity wave fluxes to those reported in this paper.

- It is not clear how equations 5 come from equations $1 - 3$, this appears to be a novel step in deriving the acoustic gravity wave dispersion relation, so would be insightful to have the steps elaborated on with their physical meaning. The reader is pointed to more detail in Jovanovic (2016), however no extra details to what is shown in this paper are given.

- Please ensure that all symbols are correctly defined. E.g., no variables in equations $(1 - 3)$ are defined except the constants.

- For brevity, it would probably be more appropriate to refer to the troposphere-stratosphere boundary as the tropopause and the stratosphere-mesosphere boundary as the stratopause. All terms appear in the paper, so I suggest picking a single convention for consistency in the paper.

- Please could the labels on the figures be made in a larger font, at least to match the font size in the paper.

- Please could you also review the spelling and grammar, as many instances of grammatical errors were spotted in the paper.

**References**

Jovanovic, G., 2016. Gravito-Acoustic Wave Reflection. Romanian Reports in Physics 68, 459–472.

---

## Author Comment (AC1)

This work considers the role of gravity waves in the evolution of the sudden stratospheric warming (SSW). In particular, the author makes use of the hydrodynamic equations to quantify the reflection coefficient of vertically propagating gravity waves (GWs) across the tropopause and stratopause during normal winter conditions compared with during an SSW in which the polar winter stratosphere warms significantly and the polar winter mesosphere cools. Consistent with previous work, the study suggests that the modification of GW flux from the troposphere to the stratosphere and from the stratosphere to the mesosphere lead to the anomalous temperatures observed in the polar stratosphere and mesosphere.

The author suggests that during an SSW, the reflection coefficient significantly increases at the tropopause, leading to significantly fewer GWs entering the stratosphere from below, while the reflection coefficient increases at the stratopause compared with the no-SSW case. It is unclear how the stratopause is being defined, and how the temperatures used to define the reflection coefficient of GWs are chosen. The study could be worthy of publication as it will likely provide new insight into how GW forcing in the stratosphere and mesosphere is modified during an SSW. The study is well written, but some things could be clarified and improved. I recommend major revisions.

**Answer 1**: The stratopause is characterized by a reversal of the atmospheric lapse rate at around 50 km ( Etienne Vignon and Daniel M. Mitchell, in Clim Dyn (2015) 44:3323–3337 DOI 10.1007/s00382-014-2292-4). It inhibits vertical exchange of air masses. However, slow vertical exchange in the stratosphere and mesosphere is present. This is mainly driven by planetary and gravity waves. I provided more details in the discussion of the article.

Also, the stratopause can be defined as in the article High Resolution Dynamics Limb Sounder observations of the gravity wave-driven elevated stratopause in 2006, by France, J. A. Et al., JOURNAL OF GEOPHYSICAL RESEARCH, VOL. 117, D20108, doi:10.1029/2012JD017958, 2012, page 2: " For this work, **we define the stratopause simply as the temperature maximum between 20 km and 90 km**."

This could be a good solution for my article although I changed the term boundary into discontinuity.

**The source of the temperature data are in:**

**U. S. Standard atmosphere**, 1976, page 10, Fig. 3.

Also, in the newest source **NRLMSIS 2.0**, ref. Emmert, J. T., Drob, D. P., Picone, J. M., Siskind, D. E., Jones, M. Jr., Mlynczak, M. G., et al. (2020). NRLMSIS 2.0: A whole-atmosphere empirical model of temperature and neutral species densities. Earth and Space Science, 7, e2020EA001321. https://doi.org/10.1029/2020EA001321, fig. 1, page 4.

In **the book "Earth's climate response to a changing Sun",** Editors: Thierry Dudok de Wit, Ilaria Ermolli, Margit Haberreiter, Harry Kambezidis, Mai Mai Lam, Jean Lilensten, Katja Matthes, Irina Mironova, Hauke Schmidt, Annika Seppälä, Eija Tanskanen, Coordinated by: Claude Bertout, Chris Biemesderfer Kleareti Tourpali, Yoav Yair, page 214- "In stratospheric applications for example is H = 7 km, representative for **a mean stratospheric temperature of 240 K**."

Also in **article How Sudden Stratospheric Warming Affects the Whole Atmosphere**, Pedatella et al. In Eos Transactions American Geophysical Union · March 2018, page 36, Fig.2 shows the temperatures in the normal atmosphere (without SSW) which are in accordance with my chosen **temperatures for troposphere/stratosphere boundary T1=220 K** (about -50° C), and for the **stratosphere/mesosphere boundary T2=270 K** (about 0° C).

**In article**- Liu, X., J. Yue, J. Xu, L. Wang, W. Yuan,J. M. Russell III, and M. E. Hervig (2014),Gravity wave variations in the polarstratosphere and mesosphere from SOFIE/AIM temperature observations,J. Geophys. Res. Atmos., 119,7368–7381, doi:10.1002/2013JD021439, on page 3, Figure 2: " (a) SOFIE

temperature profile (black) and the corresponding background temperature (red, superposition of zonalmean temperature and planetary waves with zonal wave number (WN) 1–5) at 66.6°N, 33.2°E on 1 June 2013".

Overall, the introduction needs to be reworked and additional references should be added. The first paragraph is basically one long paragraph, with the exception of the transition to Section 2. More background should be given on what drives the temperature in the polar winter stratosphere and mesosphere, both during non-SSW and SSW conditions, as these temperatures drive the changes in GW reflectivity that is the crux of the paper. In particular, there is no mention of the global residual circulation nor how GW breaking relates to the descent and warming in the stratosphere/lower mesosphere, and ascent in the middle and upper mesosphere observed during non-SSW conditions. Also, explain how modifications to GW forcing leads to the strong descent and warming in the middle and lower stratosphere.

**Answer 2:** I have written the Introduction again according to your comments. New paragraphs are:

"Depending on the phase speed of the waves and the velocity of the background wind, one can define a critical layer where the intrinsic frequency of the waves would approach the inertial frequency and the vertical wavelength would approach zero (Fritts and Alexander , 2003). If such a critical layer is present, gravity waves will break somewhere below that level and deposit more momentum already in the stratosphere. Dissipating and breaking GWs decelerate the background wind as the momentum forcing and influence planetary waves by either changing the wave guide or generating in situ planetary waves through barotropic/baroclinic instabilities (Scinocca and Zhang , 1998). Before the SSW, the stratospheric zonal mean winds are eastward. They filter out a significant portion of the eastward directed GWs, favoring the upward propagation of harmonics with phase velocities directed westward. During SSW, the deceleration of the westerly jet in the stratosphere allows more propagation of GWs with eastward phase speeds into the mesosphere, and the resultant eastward gravity wave drag (GWD) induces equatorward mass flow, resulting in the upward motion and adiabatic cooling in the polar mesosphere, (Holton , 1983; Siskind et al. , 2010; Song et al. , 2020). The unusually low temperatures at the altitude of the conventional undisturbed polar winter stratopause were linked to this reduced GWD and associated weakening of the descending branch of the mesospheric residual circulation which normally warms the winter polar stratopause (Hitchman et al. , 1989). Polar cap temperatures from the Aura Microwave Limb Sounder (MLS) averaged north of 60 $^0$ N show a joint occurrence of a warm stratosphere and a cold mesosphere in 71 percent of major warmings in 2004–2015 (Zülicke et al. , 2018). In their study, Cullens and Thurairajah (2021) analyzed 40–years of long–term ERA5 output in order to study the general trends in GWs variations before, during, and after the SSW. Their results indicate that although the main driver of SSWs are planetary waves, GWs can contribute to the occurrences and strength of SSWs."

**I used the literature listed below.**

**In article**- Increased vertical resolution in the stratosphere reveals role of gravity waves after sudden stratospheric warmings, Weather Clim. Dynam., 4, 81–93, 2023 https://doi.org/10.5194/wcd-4-81-2023 by Wolfgang Wicker, Inna Polichtchouk , and Daniela I. V. Domeisen, page 81/82 :" Typically, extra-tropical gravity waves are excited near the surface by flow over orography or in the upper troposphere by jet/front imbalances. These waves commonly propagate via the stratosphere into the mesosphere, where their amplitudes grow until the waves break. Depending on the phase speed of the waves and the velocity of the background wind, one can define a critical layer where the intrinsic frequency of the waves $\hat{\omega}$ would approach the inertial frequency f and the vertical wavelength would approach zero (e.g., Fritts and Alexander, 2003). If such a critical layer is present, gravity waves will break somewhere below that level and deposit more momentum already in the stratosphere. Compared to planetary waves, gravity waves receive less attention in extra-tropical stratosphere studies. However,

both observational and modeling studies document increased gravity wave amplitudes at the edge of the polar vortex during minor stratospheric warmings concurrently with the peak of the planetary wave flux (e.g., Duck et al., 1998; Venkat Ratnam et al., 2004; Wang and Alexander, 2009; Yamashita et al., 2010; Dörnbrack et al., 2018; Polichtchouk and Scott, 2020). During major warmings, the polar vortex breaks down and the downward propagation of the zero-wind line prevents the propagation of stationary gravity waves into the upper stratosphere and mesosphere (e.g., Wang and Alexander, 2009; Hindley et al., 2020). The absence of gravity wave breaking in the mesosphere **explains the mesospheric cooling** during an SSW by a relaxation to radiative equilibrium (Holton, 1983).

**In Cullens, C. Y. and Thurairajah, B.:** Gravity wave variations and contributions to stratospheric sudden warming using long-term ERA5 model output, Journal of Atmospheric and Solar-Terrestrial Physics, 219, 105632, https://doi.org/10.1016/j.jastp.2021.105632, 2021.,

Introduction- Although the main driver of SSWs are widely accepted as planetary waves, Albers & Birner [2014] discussed contributions of GWs to preconditioning and occurrences of SSWs. By analyzing output of GW parameterization from long-term reanalysis data, Albers & Birner [2014] reported that the contributions of GW to westward accelerations that causes SSW can be ~10-30%, and that GWs can help SSWs to occur. Vidi dolje. U Sect. 3: " **In general, GWs are larger prior to SSWs (Day -20 to Day 0) than after the wind reversal (Day 0 to Day 20),** similar to the example shown in Figure 2 and consistent with previous studies [e.g., Wang & Alexander, 2009; 166 Yamashita et al., 2010; Limpasuvan et al., 2011; Liu et al., 2017]….**GW enhancements and reductions prior and after the central day are more evident during split events than during displacement events**. … **GWs are enhanced when background winds start to weaken…** Weaker GWs during displacement events compared to the split events can be associated with the weaker background winds before wind reversal during displacement events as shown in Figure 4. Background winds are affecting GW propagation and GW filtering [e.g., Fritts and Alexander, 2003]…. In general, all hot spots of GW enhancements during split and displacement events are located at the edge of polar vortex…. Causes of GW enhancements at polar vortex edges can be preferrable propagation condition for GWs and in-situ generation of GWs due to spontaneous adjustment process…. GW enhancements seem to occur when the location of the polar vortex edge overlaps with the strong wave source, providing preferrable wave propagation from the strong tropospheric GW source region to the stratosphere. In-situ generations of GWs can also affect GW variations during SSWs and have been discussed for the 2016 minor SSW by Schoon and Zülicke, [2018] and Dörnback et al. [2018] using ECMWF data, for the 2009 major SSW by Limpasuvan et al. [2011], and by Liu [2017] using a high-resolution WACCM model…. Strongly distorted polar vortex can be a strong source of in-situ generation of GWs by adjustment process at the exit region of jet stream [Zhang et al., 2004]. Our results show that GW enhancements is located at the edge of polar vortex…. These stratospheric winds can be influenced by GWs and GWs can play an important role in pre-conditioning the atmosphere that aids SSW occurrences…. However, GWD contributed up to ~19% of the total drag when wind reversal occurred, and enhancements of westward GWD are coincident with weakening of eastward polar jet prior to SSWs from December through January…. **The major contributions of GWD are from GWs with zonal wavelengths of 100-600 km**, consistent with GW wavelengths during SSWs shown by Yamashita et al. [2010] and also Limpasuvan et al. [2011]…"

**Article**- Role of gravity waves in a vortex-split SSW in January 2009, Journal of the Atmospheric Sciences, Volume 77, Issue 10, by Song et al. 2020,

Introduction: " However, some recent studies have noted that enhanced PWs alone may not be sufficient to explain the occurrence of SSWs, and the role of GWs has emerged as an additional wave forcing to drive SSW evolution based on various observations (e.g., Whiteway et al. 1997; Duck et al. 1998, 2001; Wang and Alexander 2009), numerical modeling (e.g., Limpasuvan et al. 2011; Gavrilov et

al. 2018; Scheffler et al. 2018), and reanalysis data analyses (e.g., Albers and Birner 2014; Song and Chun 2016)….. . Cohen et al. (2014) noted that GWs play a major role in PW modulation in the winter stratosphere through changes in the waveguide of Pws…. "

**In article-** Vortex Preconditioning due to Planetary and Gravity Waves prior to Sudden Stratospheric Warmings, Journal of Atmospheric Sciences, Vol. 71, Issue 11, 2014 https://doi.org/10.1175/JAS-D-14-0026.1, by J. R. Albers and T. Birner,

page 4030: " we show that in the preconditioning phase of SSWs, both planetary waves and gravity waves play an important role in the geometric evolution of the vortex and that the majority of the combined wave forcing occurs in the region above approximately 30 km."

**In article-** AIRS satellite observations of gravity waves during the 2009 sudden stratospheric warming event. Journal of Geophysical Research: Atmospheres, 126, e2020JD034073. https://doi. org/10.1029/2020JD034073 by Kalisch, S., & Chun, H.-Y. (2021),

in Introduction: " GWs are one known candidate for vortex disruption. … From the troposphere, they propagate upwards through the middle atmosphere while gaining significant amplitude. The breaking of GWs often occurs at mesospheric altitudes and results in acceleration or deceleration of the background flow. These mechanisms are already well researched. Hence, in this paper, we focus on the impact of the background atmosphere on the distribution of GW activity in the Northern Hemisphere during the SSW event…. In conclusion: " Dissipating and breaking GWs decelerate the background wind as the momentum forcing and influence planetary waves by either changing the wave guide or generating in situ planetary waves through barotropic/baroclinic instabilities (Scinocca and Haynes, 1998)."

**In article**- The Separated Polar Winter Stratopause: A Gravity Wave Driven Climatological Feature, Journal of the Atmospheric Sciences, Vol. 46, Issue 3, page: 410-422, https://doi.org/10.1175/1520-0469(1989)046<0410:TSPWSA>2.0.CO;2 by Matthew H. Hitchman, John C. Gille, Clive D.Rodgers, and Guy Brasseur, on page 411: " These calculations support the hypothesis that gravity wave absorption induces a mesospheric flow toward the winter pole, with the descending branch of the circulation bringing high entropy air downward over the pole in a confined region. This downward advection maintains elevated temperatures against radiation to space... On page 412:" **Gravity waves will break when their amplitudes are so large that they become convectively unstable.** "

**In article**- Körnich, H., and E. Becker (2010), A simple model for the interhemispheric coupling of the middle atmosphere circulation,*Adv. Space Res., 45,* 661–668, https://doi.org/10.1016/j.asr.2009.11.001, on page 3: "This flow (it is the mean circulation) corresponds approximately to the mean trajectories of air parcels (Dunkerton, 1978) and **it is mainly driven by zonal forces from the breaking of gravity and planetary Rossby waves, the so-called 'wave drag'** (e.g. Holton and Alexander, 2000).

**In article**- Stephan, C. C., Schmidt, H., Zuelicke, C., and Matthias, V.: Oblique gravity wave propagation during sudden stratospheric warmings, Journal of Geophysical Research: Atmospheres, 125, e2019JD031528, https://doi.org/10.1029/2019JD031528, 2020,

Introduction: " Gravity waves (GWs) are the major contributor to the mesospheric cooling that has been observed to accompany stratospheric warming (Quiroz, 1969; Labitzke, 1972). Polar cap temperatures from the Aura Microwave Limb Sounder (MLS) averaged north of 60 ◦ N show a joint occurrence of a warm stratosphere (10 hPa) and a cold mesosphere (0.01 hPa) in 71% of major warmings in 2004–2015 (Zülicke et al., 2018). The causal chain for this stratosphere-mesosphere coupling is well understood: under undisturbed winter conditions GWs propagate through stratospheric westerlies, which filter the GW pseudo-momentum flux (GWMF) to be predominantly westward in the

mesosphere. The westward drag induces a northward and downward mesospheric residual circulation, which dynamically controls the winter stratopause (Holton, 1983). In contrast, when stratospheric winds weaken or change to easterly directions, the mesospheric GW forcing results in less warming or even a cooling."

**Article**- Role of gravity waves in vertical coupling during sudden stratospheric warmings, Yiğit and Medvedev, Geosci. Lett. (2016) 3:27 DOI 10.1186/s40562-016-0056-1, Introduction: "Sudden stratospheric warmings (SSWs) are spectacular events that **disturb the circulation in the winter hemisphere.**"

On page 3: "In the middle atmosphere, the main mechanism of GW obliteration is nonlinear breaking and/or saturation that occurs when amplitudes become large."

On page 7: "Namely, before the warming, the stratospheric zonal mean winds are eastward. They filter out a significant portion of the eastward directed GWs, favoring the upward propagation of harmonics with phase velocities directed westward. During the warmings, the decelerating westerlies increase the chances of GWs with eastward horizontal phase speeds to propagate to higher altitudes (Yiğit and Medvedev 2012).... **The deceleration of the stratospheric eastward zonal flow during sudden warmings leads, ultimately, to an upward circulation in the mesosphere that results in mesospheric cooling.**"

**In article**- Siskind, D. E., S. D. Eckermann, J. P. McCormack, L. Coy, K. W. Hoppel, and N. L. Baker (2010), Case studiesof the mesospheric response to recent minor, major, and extended stratospheric warmings, J. Geophys. Res., 115, D00N03,doi:10.1029/2010JD014114,

on page 1: " Holton [1983] pointed out that **the transition to stratospheric easterlies will act to reduce, or eliminate, the flux of tropospheric gravity wave momentum entering the mesosphere, leading to cooling** via reduced wave-induced diabatic descent [Garcia and Boville, 1994].

On page 2: " The unusually low temperatures at the altitude of the conventional undisturbed polar winter stratopause were linked to this reduced orographic gravity wave drag and associated weakening of the descending branch of the **mesospheric residual circulation** which normally warms the winter polar stratopause[Hitchman et al., 1989]."

On page 14: " One consistent pattern that emerges from this study is the reduction and then **disappearance of parameterized OGWD in the upper stratosphere and mesosphere as the stratosphere becomes progressively more disturbed. As the stratospheric westerly zonal flow progressively weakens or reverses, orographic gravity waves increasingly encounter stratospheric critical lines where they are completely absorbed. This removal of OGWD forcing from the stratosphere and mesosphere serves to remove a source of wave-driven mean downwelling leading to cooling at 50 km.** "

**In article**- Role of Gravity Waves in a Vortex-Split Sudden Stratospheric Warming in January 2009, by Song, B. G., Chun, H. Y., and Song, I. S., Journal of the Atmospheric Sciences, Volume 77, Issue 10, p.3321-3342, 10.1175/JAS-D-20-0039.1, 2020,

on page 3324: " Although the period of sudden increase in the temperature anomalies generally coincides with that of a sudden decrease in the PNJ, the temperature increase starts earlier (from 10 January) in a shallow layer near z =50 km, just above the cold anomalies between z = 20 and 40 km. The temperature above approximately 55 km is cold throughout the whole considered period, which becomes even colder after lag =0….. . The deceleration of the westerly jet in the stratosphere during

SSWs allows more propagation of GWs with eastward phase speeds into the mesosphere, and the resultant eastward GWD induces equatorward mass flow, resulting in the upward motion and adiabatic cooling in the polar mesosphere.

**In article-** Meriwether, J. W., and A. J. Gerrard (2004), Mesosphere inversion layers and stratosphere temperature enhancements, *Rev. Geophys.*, 42, RG3003, doi:10.1029/2003RG000133. ,

on page 4: "… at the bottom side of the thermal layers (where the temperature profile is increasing in altitude) the positive temperature gradient with increasing altitude signifies an increase in atmospheric stability and reduction of vertical mixing. In contrast, at the topside of the thermal layer the negative temperature gradient with decreasing altitude implies a reduction in atmospheric stability to the point that the **atmosphere may become convectively unstable**, thus possibly supporting the development of turbulence. " …This topside instability can be an additional, and currently unaccounted for, in situ source of such gravity waves. …On page 8: "Holton [1983] noted **that gravity wave forcing was a primary contributor to the stratopause** and mesosphere **warmin**g."

On page 9: "Gerrard et al. [2002] highlighted the thermal structure of the polar vortex throughout the entire middle atmosphere. They found that in the vortex core, there was a warm upper stratosphere and a cold lower stratosphere. As the lidar observatory position changed from a point inside the vortex core to a point within the vortex jet (and eventually outside of the vortex altogether), temperatures in the upper stratosphere became cooler and temperatures in the lower stratosphere were warmer."

**Article-** Ern, M., Trinh, Q. T., Kaufmann, M., Krisch, I., Preusse, P., Ungermann, J., Zhu, Y., Gille, J. C., Mlynczak, M. G., Russell III, J. M., Schwartz, M. J., and Riese, M.: Satellite observations of middle atmosphere gravity wave absolute momentum flux and of its vertical gradient during recent stratospheric warmings, Atmos. Chem. Phys., 16, 9983–10019, https://doi.org/10.5194/acp-16-9983-2016, 2016.

In abstract. : "Sudden stratospheric warmings (SSWs) are circulation anomalies in the polar region during winter…. Both planetary waves and gravity waves contribute to the onset and evolution of SSWs." On page 9983: "**The unperturbed arctic winter stratosphere** is characterized by a strong eastward-directed zonal wind jet (polar vortex). Occasionally, however, forcing by upward propagating planetary Rossby waves can lead to strong deceleration and even reversals of this polar jet. These events are associated with strong warming of the polar stratosphere, and they are therefore called sudden stratospheric warmings (SSWs). Page 9984: "Particularly during major warmings, stationary **planetary waves will encounter critical levels when the zonal jet reverses from eastward to westward.** The waves dissipate and can no longer propagate to higher altitudes. However, as was shown by Holton (1983), accurate **representation of major SSWs also requires the inclusion of gravity wave drag in models**."… ". Effects of SSWs are not limited to the polar region. **SSWs influence the global meridional residual circulation, and meridional coupling between different latitudes is observed**. For example, SSWs have influence on mesospheric temperatures in the tropics (e.g., Shepherd et al., 2007), and they likely also have an effect on the opposite hemisphere (e.g., Becker and Fritts, 2006; de Wit et al., 2015; Cullens et al., 2015)."…. "A recent review on the mechanism **of downward coupling during SSWs** is given, for example, by Kidston et al. (2015)"

Page 9985: " …the effect of **SSWs is not limited to the stratosphere**. **Also the mesosphere** (e.g., Labitzke, 1972; Jacobi et al., 2003; Siskind et al., 2005, 2010; Hoffmann et al., 2007; Yamashita et al., 2013; Zülicke and Becker, 2013), **and even the thermosphere/ionosphere are affected** (e.g., Goncharenko and Zhang, 2008; Fuller-Rowell et al., 2010; Yigit et al., 2014). **The selective filtering of**

**gravity waves by the anomalous winds during (major) SSWs is an important mechanism in these vertical influences.** SSWs are associated with **mesospheric coolings** (e.g., Labitzke, 1972; Siskind et al., 2005; Hoffmann et al., 2007) **that are likely driven by dissipation of eastward propagating gravity waves** (e.g., Holton, 1983; Miller et al., 2013). **As a consequence, the zonal wind in the mesosphere/lower thermosphere** (MLT), **that is usually directed westward during winter, can change its sign to eastward.** …. Gravity waves could also play an important role for the onset and triggering of SSWs. As has been pointed out by **Albers and Birner (2014),** before the onset of a SSW, gravity wave drag in the stratosphere is non-negligible. Therefore, gravity waves may contribute to the preconditioning of the polar vortex and its shape such that resonant amplification of planetary wave amplitudes occurs and a SSW takes place."

On page 9986: " Other studies revealed that sometimes after major SSWs, gravity wave activity in the stratosphere is strongly suppressed (e.g., Wright et al., 2010; Thurairajah et al., 2014). On the other hand, before the major SSW in winter 2005/2006, an enhancement of gravity wave momentum flux was observed in the lower mesosphere (France et al., 2012)."

On page 1992: " **There are two main processes related to the background wind that shape the gravity wave distribution. The first process is critical level filtering:** during wave propagation it can happen that background wind u and ground-based phase speed c of a gravity wave become equal. In this case, the intrinsic phase speed of the wave cf=c-u becomes zero, **the wave cannot propagate further, and it dissipates completely**. **The second process is wave saturation**. If a gravity wave propagates conservatively upward (without dissipation), **the wave amplitude will grow exponentially** according to the decrease in background density. At some point, the wave amplitude cannot grow further**. The wave amplitude reaches its saturation limit, and the wave breaks**. This can happen without a critical level being reached. The saturation temperature amplitude is proportional to the intrinsic phase speed, i.e., to the difference between ground-based phase speed and background wind. **The saturation momentum flux of a gravity wave is proportional to the intrinsic phase speed to the power of 3.**"... **During these unperturbed periods**, on zonal average here is no wind reversal in the stratosphere and lower mesosphere. Under these conditions, gravity waves with westward or zero ground-based phase speeds can propagate in this whole-altitude range without encountering critical levels. This means that those gravity waves can attain large amplitudes already in the stratosphere and lower mesosphere, because their intrinsic phase speeds and thus their saturation temperature amplitudes are high..... **Compared to the situation of strong polar vortices, during weak vortex conditions,** gravity wave squared amplitudes in the midstratosphere around 30 km altitude are somewhat reduced. A likely reason for this reduced gravity wave activity are reduced gravity wave saturation amplitudes that are not enhanced by strong favorable background winds..... **the highest values of gravity wave squared amplitudes** (about 10K2 and more) in the mid stratosphere (at _30 km altitude) are found before or around the central date of strong major SSWs with PJO event."

On page 1994: " **After the onset of SSWs** associated with PJO events, **gravity wave activity in the stratosphere is reduced for two reasons. First**, zonal winds are much weaker, not resulting in favorable enhancements of gravity wave saturation amplitudes. **Second**, due to anomalous westward winds there are wind reversals in the troposphere and/or in the stratosphere, such that gravity waves with zero ground-based phase speed (e.g., mountain waves) or with slow westward-directed phase speeds will encounter critical levels."

On page 1996: " **Much of the discussion of gravity wave squared amplitudes** in Sect. 4.1 **is also valid for gravity wave momentum fluxes** and will therefore not be repeated in detail. The main difference is that gravity wave amplitudes usually grow with altitude. If a gravity wave propagates conservatively in a constant wind, this amplitude growth is exponential, compensating the exponential decrease of atmospheric background density with altitude. Different from this, gravity wave pseudomomentum flux is conserved, i.e., remains constant, if a wave propagates conservatively. In all panels of Fig. 5, however, we find that gravity wave momentum flux gradually decreases with altitude, indicating an overall dissipation of gravity waves while the waves are propagating upward."

On page 1999: " The absence of strong gravity wave drag on the lower flank of the jet is in good agreement with the theoretical picture that **the residual circulation drives the thermal structure of the mesosphere and the stratopause in the polar region**, and the new polar jet is forming in response to these changes in the residual circulation: in the mesosphere, **the gravity-wave-driven branch of the residual circulatio**n, which is directed poleward and downward in the polar region, enforces the warm winter stratopause (e.g., Hitchman et al., 1989). During SSWs, anomalous breaking of planetary waves changes the circulation in the stratosphere, and, as a consequence, the net forcing by gravity waves changes its sign, which leads to an **anomalous residual circulation** resulting in a cooling of the (upper) stratosphere and mesosphere (i.e., at altitudes above about 50 km). Later, during the jet recovery, the sign of net gravity wave forcing changes again, and the stratopause is rebuilt (e.g., Tomikawa et al., 2012; Hitchcock and Shepherd, 2013). The theoretical picture of the mesospheric gravity-wave-driven branch of the residual circulation being responsible for changes in the residual vertical motion and related dynamical warming is well supported by the fact that the strongest gravity wave potential drag is usually observed above the temperature maximum of the stratopause (cf. Figs. 2 and 6).

On page 10013: " In particular, **our findings support the study by Albers and Birner (2014), and it is suggested that gravity waves may contribute to the triggering of SSWs by preconditioning the shape of the polar vortex such that a SSW can take place. ....** Only gravity waves **with horizontal wavelengths longer than about 100–200 km are visible** for those instruments**."**

Page 7: " **The mesospheric meridional circulation** which consists of summer-to-winter meridional flow and descent (ascent) motion around the winter (summer) pole **is primarily driven by momentum deposition due to gravity waves** [Holton, 1983; Garcia and Solomon, 1985]. **A significant role of the GWs in driving the meridional circulation above 0.1 hPa is commonly observed both before and after the SSW in this GCM**.

**In article-** Liu, X., J. Yue, J. Xu, L. Wang, W. Yuan, J. M. Russell III, and M. E. Hervig (2014), Gravity wave variations in the polar stratosphere and mesosphere from SOFIE/AIM temperature observations, J. Geophys. Res. Atmos., 119, 7368–7381, doi:10.1002/2013JD021439, in Introduction: " Gravity waves (GWs) play important roles in influencing the circulations and structures of the atmosphere by transporting their momentum and energy from the lower to the upper atmosphere [Lindzen, 1981; Holton, 1983; Dunkerton, 1997; Fritts and Alexander, 2003; Alexander et al., 2010]. **When GWs break, the resultant momentum deposition drives a residual circulation from the summer to winter mesosphere. This circulation is upwelling in the summer mesosphere, causing adiabatic cooling in the summer polar mesopause region where the atmospheric state is far away from the radiative equilibrium** [Leovy, 1964; Holton, 1983; Andrews et al., 1987]. **The cold summer mesopause provides a favorable environment for polar mesospheric clouds (PMCs) to form** [Rapp and Thomas, 2006; Hervig et al., 2009a, 2009b, 2013]. **Moreover, GWs in the polar regions and their variations influence the brightness and backscattering of PMCs** [Hines, 1968; Gerrard et al., 2004; Thayer et al., 2003; Chu et al., 2009; Chandran et al., 2010]. **Thus, it is important to study the climatology of GWs in these regions**."

**In article**- Kuilman, M. S. and Karlsson, B.: The role of the winter residual circulation in the summer mesopause regions in WACCM, Atmos. Chem. Phys., 18, 4217–4228, https://doi.org/10.5194/acp-18-4217-2018, 2018, in Introduction: "**The circulation in the mesosphere is driven by atmospheric gravity waves** (GWs). These waves originate from the lower atmosphere and as they propagate upwards, **they are filtered by the zonal wind in the stratosphere** (e.g. Fritts and Alexander, 2003). Because of the decreasing density with altitude and as a result of energy conservation, the waves grow in amplitude. At certain altitudes, the waves – depending on their phase speeds relative to the background wind – **become unstable and break**. **At the level of breaking**, the waves deposit their momentum into the background flow, **creating a drag on the zonal winds in the mesosphere, which establishes the pole-to-pole circulation** (e.g. Lindzen, 1981; Holton, 1982, 1983; Garcia and Solomon, 1985). **This circulation drives the temperatures far away from the state of radiative balance, by adiabatically heating the winter mesopause and adiabatically cooling the summertime mesopause** (Andrews et al., 1987; Haurwitz, 1961; Garcia and Solomon, 1985; Fritts and Alexander, 2003)… On page 4218: "**Due to the eastward zonal flow in the winter stratosphere, GWs carrying westward momentum propagate relatively freely up into the mesosphere where they break.** Therefore, in the winter mesosphere, the net drag from GW momentum deposition is westward. When vertically propagating PWs break – also carrying westward momentum – in the stratosphere, its momentum deposited onto the mean flow, which decelerates the stratospheric westerly winter flow. To put it short, a weaker zonal stratospheric winter flow allows for the upward propagation of more GWs with an eastward phase speed, which, as they break, reduces the westward wave drag (see Becker and Schmitz, 2003, for a more rigorous description). **This filtering effect of the zonal background flow on the GW propagation results in a reduction in strength of the winter-side mesospheric residual circulation** when the BDC is stronger. **This weakened meridional flow causes the mesospheric polar winter region to be anomalously cold and the tropical mesosphere to be anomalously warm** (Becker and Schmitz, 2003; Becker et al., 2004; Körnich and Becker, 2009)."

My other significant concern is in regards to the temperatures you are using for the tropopause/stratosphere/stratopause to define the parameter *s*, which drives the primary findings of this work. Where are these temperature values coming from, and what latitude/altitude are they referencing? Maybe more confusing is that you use a single temperature to represent each of these, and especially so for the stratosphere given the broad range of temperatures typically found there).

**Answer 3:** For the first part of your comment regarding the temperatures, see answer 1.

**About a single temperature**-  I used the same approach as in the article by Nina, A. and Čadež, V.: Detection of acoustic-gravity waves in lower ionosphere by VLF radio waves, Goephysical Research Letters, 40, 18, 4803-4807, https://doi.org/10.1002/grl.50931, 2013, page 4805: "**This is shown in Figure 3 taking 250 K as typical temperature of the ionosphere below 90 km.** "

**Also, in article** Horizontal and vertical propagation and dissipation of gravity waves in the thermosphere from lower atmospheric and thermospheric sources by Sharon L. Vadas, JOURNAL OF GEOPHYSICAL RESEARCH, VOL. 112, A06305, doi:10.1029/2006JA011845, 2007, on page 2: "**Although the average temperature in the lower atmosphere is T ~ 250 K**, the temperature increases rapidly in the lower thermosphere. **During extreme solar minimum, the thermosphere is relatively cold, T~600 K. During active solar conditions however, the temperature in the thermosphere can be T~2000 K".**

**Article**-Phases and amplitudes of acoustic-gravity waves II: The effects of reflection, Astronomy and Astrophysics. 278, 617-626 (1993) by Marmolino el al. in abstract: " **We study wave reflection caused by the temperature stratification of the solar atmosphere, assumed to be a succession of**

**two layers of different temperatures…**" Page 620, Fig. 2: "Amplitudes and phases of the reflection and transmission coefficients R and T vs. frequency at $k_x$=1.31 Mm$^{-1}$. Solid lines refer to **a succession of two layers suitable to represent the photospheric stratification** ($T_l$=**5000 K** (this means T in the lower layer); $T_u$=**4500 K** (this means T in the upper layer)…"  The same is in my article where the succession of the two layers with temperatures $T_1$=220 K and $T_2$=240 K represents the atmospheric stratification between the upper troposphere/lower stratosphere and middle stratosphere; also the temperatures  $T_2$=240 K and $T_3$=270 K represent the stratification between the middle stratosphere and the upper stratosphere/lower mesosphere.

**In article:** Acoustic-gravity wave propagation characteristics in 3D radiation hydrodynamic simulations of the solar atmosphere, in Philosophical transactions of the royal society A by Fleck et al. https://doi.org/10.1098/rsta.2020.0170, page 2. : "A theoretical phase difference spectrum for an isothermal atmosphere with **sound speed 6.5 km s$^{-1}$** following [Souffrin P. 1972 Radiative relaxation of sound waves in an optically thin isothermal atmosphere. Astron. Astrophys. 17, 458] is shown in figure 2 in black…" Also, page 3, Figure 3: "Observed phase difference spectra between Fe 6302 and Fe 6301 derived from two Hinode SP sit-and-stare runs in the quiet Sun. The parameters used for the theoretical curve (solid line) are /span>z = 60 km, **cs = 7.5 km s$^{-1}$** , $\tau$R = 40 s and kh = 1.25 Mm−1." Same article - figures 7, 8, 9, 10 **refer to the solar isothermal atmosphere**.

**Article**- THE ENERGY FLUX OF INTERNAL GRAVITY WAVES IN THE LOWER SOLAR ATMOSPHERE in The Astrophysical Journal, 681: L125–L128, 2008 July 10, by Straus et al. 2008, on page L127 : "Using **7.5 km s$^{-1}$ for the speed of sound**, obtained from a fit of the high-frequency part of the V-V phase spectrum F($k_{II}$, q) between the two heights in the IBIS data set to a theoretical spectrum following the theory of linear waves (Souffrin 1966), and $\gamma$=5/3, we obtain $\omega_{BV}$= 4.75 mHz."

The constant sound speed in these two articles means the constant atmospheric temperature because $v_s$=( $\gamma$RT)$^{1/2}$.

**Article**- Observed Local Dispersion Relations for Magnetoacoustic-gravity Waves in the Sun's Atmosphere: Mapping the Acoustic Cutoff Frequency, The Astrophysical Journal Letters, 884:L8 (5pp), 2019 October 10,  DOI 10.3847/2041-8213/ab4719, by Stuart el al., on page 2: " using Equation (2) and kz as defined by the dispersion relation for acoustic-gravity waves in an **isothermal stratified atmosphere** with constant radiative damping..."

On line 150, you say that the stratosphere temperature can rise from 240 K to 290 K during an SSW. Do you have a reference for this? Manney et al. (2008) and France et al. (2012a, b) show the highest stratospheric temperatures during an SSW to be ~280 K. Also, polar-cap-mean temperature anomalies associated with SSWs are only on the order of 20 K (Vignon and Mitchell (2015). A 50 K increase in temperature is only valid if you consider the "maximum temperature anomaly (that occurs within the range of 30–90° latitude and 300 to 1hPa)" (Butler et el., 2017).

**Answer 4:** The references for the temperature increase of 50 K are:

**Article**- Role of Gravity Waves in a Vortex-Split Sudden Stratospheric Warming in January 2009, Song et al. 2020, Journal of the Atmospheric Sciences, Volume 77, Issue 10, page 3 in Results: " **At lag=0, the temperature in the polar region is 50 K higher** than that at lag =-9." Here, a central date of the SSW i.e. lag=0. Also in Summary and discussion, page 18: "As the polar vortex breaking evolves, cold air confined within the vortex is split into two areas, while warm air intrudes into polar regions, **which suddenly increases the polar air temperature by more than 50 K**."

**Article**- Role of gravity waves in vertical coupling during sudden stratospheric warmings, Yiğit and Medvedev in Geosci. Lett. (2016) 3:27, page 6: "Within about 5 days, the zonal mean temperature at 10

hPa increases **by more than 60 K** (from 200 to more than 260 K) at the North Pole, that is, more than 30 % increase (top panel)."

**Article-** Case studies of the mesospheric response to recent minor, major, and extended stratospheric warmings, JOURNAL OF GEOPHYSICAL RESEARCH, VOL. 115, D00N03, doi:10.1029/2010JD014114, 2010 by Siskind et al., in Introduction: "The largest SSWs can warm the polar winter stratosphere **by as much as 60 K or more** over a period of about a week [Labitzke, 1981].

**Article-** Effects of Sudden Stratospheric Warming Events on the Distribution of Total Column Ozone over Polar and Middle Latitude Regions, in Open Journal of Marine Science, Vol. 6 No. 2, 2016, by Madhu, V. in Abstract: "During the Sudden Stratospheric Warming events, the polar stratospheric temperature rises concurrently zonal-mean zonal flow weakens over a short period of time. As the zonal flow weakens, the stratospheric circulation becomes highly asymmetrical and the stratospheric polar vortex is displaced off the pole. **The polar stratospheric temperature rises by 50°C** and the stratospheric circumpolar flow reverses direction in a span of just few days.

**Article-** A sudden stratospheric warming compendium, Earth Syst. Sci. Data, 9, 63–76, 2017 www.earth-syst-sci-data.net/9/63/2017/doi:10.5194/essd-9-63-2017 by Butler et al. on page 72: "On average, **the maximum temperature anomaly of ∼ 50 K peaks** 1–2 days prior to the zonal wind reversal (Fig. 7a, bold black line), **but the amplitude** and timing **vary substantially among the individual events** (colored lines), **with values from 10 to almost 100 K**. Likewise, the mean latitude where the temperature maximizes tends to fall between 60 and 70° N (Fig. 7b) but ranges from ∼ 45° N to the pole."

**Article-** How Sudden Stratospheric Warming Affects the Whole Atmosphere, Pedatella et al. In Eos Transactions American Geophysical Union · March 2018. In Introduction, page 35: "During SSWs, stratospheric temperatures can fluctuate **by more than 50°C** over a matter of days."

**Article-** AIRS Satellite Observations of Gravity Waves During the 2009 Sudden Stratospheric Warming Event in Journal of Geophysical Research: Atmospheres, 126, e2020JD034073. https://doi. org/10.1029/2020JD034073 by Kalisch, S., & Chun, H.-Y. (2021), on page 14: "The major SSW of January 2009 brought record-breaking wind reversal and unprecedented high stratospheric temperatures into the Arctic region. As a result, the polar vortex had been split apart, enabling subarctic air masses to enter the polar region and further increase stratospheric temperatures **up to 50 K** above the long-term average…... Our observations are based on stratospheric GWs, as the AIRS GW retrievals are most sensitive between 25 km and 40 km."

Nevertheless, I will change the sentence in line 17/18 in: "This is a rapid warming with the temperature increase of several tens of degrees in just a few days."

 This is in agreement with Stephan et al. in article Oblique Gravity Wave Propagation During Sudden Stratospheric Warmings, published in Journal of Geophysical Research: Atmospheres, 125, 2020, e2019JD031528. https://doi.org/10. 1029/2019JD031528, Introduction, page 1: "Sudden stratospheric warmings (SSWs) are marked by a sharp **temperature increase of several tens of kelvins** in the middle to upper stratosphere (30–50 km)- over the course of several days (Butler et al., 2015)" and with Rupp, P., Spaeth, J., Garny, H., and 335 Birner, T.: Enhanced polar vortex predictability following sudden stratospheric warming events, Geophysical Research Letters, 50, e2023GL104057, https://doi.org/10.1029/2023GL104057, 2023. In Introduction: " ….with **stratospheric polar temperatures increasing by several tens of degrees** over the course of a few days."

Also, I decided to take into account both-the stratospheric warming and mesospheric cooling in a way that the stratospheric temperature will increase for 25 K, from the $T_1$=240K to $T_1^{'}$=265K and the

temperature in the mesosphere will decrease for 25 K, from the $T_2$=270K to $T_2'$=245K, i.e. the total temperature change is 50 K. This is in agreement with Limpasuvan, V., Y. J. Orsolini, A. Chandran, R. R. Garcia, and A. K. Smith (2016), in **article** - On the composite response of the MLT to major sudden stratospheric warming events with elevated stratopause, J. Geophys. Res.Atmos., 121, 4518–4537, doi:10.1002/2015JD024401, Figure 3, page 4522;

**also with C. Y. Cullens and B. Thurairajah in article** Gravity wave variations and contributions to stratospheric sudden warming using long-term ERA5 model output and references therein, Journal of Atmospheric and Solar-Terrestrial Physics, 219, 105632, https://doi.org/10.1016/j.jastp.2021.105632, 2021, Introduction: " **During major SSW events,** zonal-mean zonal winds at 60°N, 10 hPa reverses from eastward to westward **and zonal-mean temperature at 80°N, 10 hPa increases by more than 20 K** [Charlton and Polvani, 2007; Butler et al., 2017]….These temperature changes are due to changes in gravity wave (GW) and/or in planetary waves (ref.). " Also in article- Role of Gravity Waves in a Vortex-Split Sudden Stratospheric Warming in January 2009, by Song et al, Introduction: " ...the event that occurred on 24 January 2009 (hereafter SSW09) was the strongest, **with temperatures higher than 265 K over the polar regions** (Manney et al. 2009; Harada et al. 2010);"

**also with** Yamazaki, Y., Matthias, V., Miyoshi, Y., Stolle, C., Siddiqui, T., Kervalishvili, G., et al. (2020), September 2019 - Antarctic sudden stratospheric warming: Quasi-6-day wave burst and ionospheric effects. Geophys. Res. Lett.Geophysical Research Letters, 47, e2019GL086577, https://doi.org/10.1029/2019GL086577, on page 1: "According to the definition by the World Meteorological Organization (McInturff, 1978), a "minor" SSW occurs when a large temperature increase is observed in the winter polar stratosphere, **at least by 25 K** in a week or less.

**Finally**, in Limpasuvan, V., Y. J. Orsolini, A. Chandran, R. R. Garcia, and A. K. Smith (2016), On the composite response of the MLT to major sudden stratospheric warming events with elevated stratopause, J. Geophys. Res. Atmos., 121, 4518–4537, doi:10.1002/2015JD024401, on page 4521: "Above the descended stratopause (and surrounding warm layer), the atmosphere up to 100 km undergoes **tremendous cooling (by ~30 K** compared to climatology, as shown below)..."

Another concern is that it is unclear how you define the stratopause. On Line 172 you say, "During the SSW, the stratospheric temperature rises to 290 K, causing a change in the parameter s =T1/T2, which becomes 1.1". This implies that the stratosphere is warmer than the stratopause. Typically, the stratopause is defined as the layer of highest temperature, so during an SSW, the stratopause descends with the warming. Maybe you could consider referring to specific altitudes in your analysis, like 20 km, 40 km, and 50 km.

**Answer 5:** I have changed the article in Sections 4 and 5 according to your suggestions. I include the altitudes together with their temperatures: **Sect. 4.1**-" The reflection coefficient for gravity waves traveling from the upper troposphere/lower stratosphere, where the temperature is approximately 220 K at an altitude of 20 km, to the middle stratosphere, characterized by a temperature of 240 K at an altitude of 35 km, is presented in Fig. 2. The specified temperatures illustrate the temperature stratification within the stratosphere from its lower to middle region, that is, from an altitude of about 20 km to an altitude of about 35 km, (U.S. Standard Atmosphere , 1976; Liu et al. , 2014; Emmert et al. , 2020). Here, the parameter s = T 1 /T 2 has the value of s = 0.91…."

**Sect. 4.2**- Gravity waves originating in the stratosphere can travel to the mesosphere. At normal atmospheric conditions, the temperature in the middle stratosphere, at an altitude of about 35 km, is T 1 = 240 K, while the temperature in the upper stratosphere/lower mesosphere, at an altitude of about 55 km, is T 2 = 270 K, (U.S. Standard Atmosphere , 1976; Liu et al. , 2014; Emmert et al., 2020). These temperatures, which effectively demonstrate the temperature stratification within the stratosphere from its mid to upper region, yield a parameter s value of s = T 1 /T 2 = 0.89. During the SSW, the

temperature in the middle stratosphere, at an altitude of about 35 km, rises from 240 K to T 1 = 265 K, while the temperature in the upper stratosphere/lower mesosphere, at an altitude of about 50 km, decreases from 270 K to T 2 = 245 K, (Siskind et al. , 2010; Limpasuvan et al. , 2016) causing a change in the parameter s = T 1 /T 2 , which becomes s = 1.1. This changes the conditions for GWs propagation.”

**In the discussion I commented on the descent of the stratopause.**

“ The stratopause is the boundary between the stratosphere and the mesosphere at an altitude of about 55 km (Song et al. , 2020; Okui et al. , 2024). It is characterized by a reversal of the atmospheric lapse rate (Vignon and Mitchell , 2015). The beginning of the SSW is characterized by the rapid descent of the stratopause and surrounding warm layer into the stratosphere, associated with warming that is characteristic of SSW. The stratopause reaches its lowest altitude at around 30 km (Ern et al. , 2016). Above the descended stratopause, the atmosphere experiences a dramatic cooling of about 30 K at an altitude of 50 km, parallel to stratospheric warming (Limpasuvan et al. , 2016; Siskind et al. , 2010). In this article, the stratopause is assumed to be a plane boundary between the stratosphere and the mesosphere. Its altitude is not relevant for the results obtained in the analysis, since the results depend only on the temperature ratio, i.e. depend on the values of the parameter s. These values are computed assuming a temperature increase of 25 K in the middle stratosphere at an altitude of about 35 km, and a temperature decrease of 25 K in the lower mesosphere at an altitude of about 50 km. This is in accordance with aforementioned scientific literature.”

**I have used the articles listed below.**

**In the article**- The stratopause evolution during different types of sudden stratospheric warming event by Etienne Vignon · Daniel M. Mitchell, published in Clim Dyn (2015) 44:3323–3337 DOI 10.1007/s00382-014-2292-4, on page 3327: “ All three data sets show that, during an SSW event, a warm temperature anomaly (red) quickly propagates downward from the upper winter stratosphere to the mid-low stratosphere. Subsequently the polar stratosphere becomes nearly isothermal and the normal winter stratospheric state reappears a few days later. The stratopause (green line) follows well the quick downward propagation of warm anomalies and abruptly drops during the events.”

**In article**-Chandran, A., R. L. Collins, R. R. Garcia, and D. R.Marsh (2011), A case study of an ele-vated stratopause generatedin the Whole Atmosphere Community Climate Model, Geophys.Res. Lett., 38, L08804, doi:10.1029/2010GL046566,

on page 1: “The major SSW in the northern hemisphere (NH) winter of 2005/2006 was one of the strongest observed [Siskind et al., 2007; Manney et al., 2008]. **The stratopause warmed and de-scended by ~30 km** in the third week of January 2006. **This warming was accompanied by meso-spheric cooling**. This SSW was followed by a breakdown of the polar vortex and formation of an ill-defined, **almost isothermal and cooler stratopause**. ... Siskind et al. [2007, 2010] have shown that there is a reduction of GW drag in the upper stratosphere and mesosphere following the SSW. Wang and Alexander[2009] have documented significantly reduced GW amplitudes in the lower mesosphere and increased GW amplitudes in the stratosphere during SSW events. They attributed this to the exis-tence of GW critical levels (where the background wind speed is the same as GW phase speed) near the stratopause, which filters out the propagation of mainly orographic GW into the mesosphere. Wright et al. [2010] have also documented wind filtering of gravity waves during SSW events with reduction of the gravity wave momentum flux in the stratosphere. However in WACCM simulations of a major stratospheric warming and elevated stratopause, **we find that the net gravity wave forcing in the mesosphere is reversed from westward to eastward, due to the action of eastward propagating non-orographic gravity waves penetrating into the mesosphere. These non-orographic GWs play**

**a crucial role in reversing the mesospheric jet and the residual circulation thereby leading to the reformation of the stratopause in the lower mesosphere**.

On page 4: " In MY 1973–1974 there are two periods of enhanced poleward flow in the upper stratosphere (∼40–60 km) and equatorward flow in the mesosphere (∼70–90 km) that start on December 12th and December 21st. During these periods there is enhanced downward flow in the stratosphere and upward flow in the mesosphere, and warming in the stratosphere and cooling in the mesosphere."

**In article**- The roles of planetary and gravity waves during a major stratospheric sudden warming as characterized in WACCM, V. Limpasuvan, J. H. Richter, Y. J. Orsolini, F. Stordal, O.-K. Kvissel, Journal of Atmospheric and Solar-Terrestrial Physics, Volumes 78–79, 2012, Pages 84-98, https://doi.org/10.1016/j.jastp.2011.03.004,

on page 84:"... as **the stratopause** (formed initially near its climatological position) **descends 10–20 km toward** the middle stratosphere with SSW onset….. On page 94/95: " A strong interplay between GW and PW forcing is evident in the simulation as the middle atmosphere undergoes a major SSW and eventually recovers. Climatologically, westward GWD drives the polar atmosphere above the 45 km in promoting poleward and downward motion over the region throughout the winter. This forcing helps to maintain a warm stratopause layer centered around 60 km and keeps the polar night jet weaker than it would be otherwise due strictly to radiative cooling (Hitchman et al., 1989). When the winter circumpolar wind becomes anomalously weak, EP flux due to PWs persistently converges in that weakened wind region to initiate SSW. This flux convergence provides a strong westward forcing that leads to an extensive area of wind reversal (to the westward direction) in the polar mesosphere and the upper stratosphere. The lower boundary of the resulting westward wind pocket (marked by the zero-wind surface) then serves as focal area for further interactions between PW and the mean flow and descends in time toward the lower stratosphere. The descent is marked by the falling stratopause layer along the zero-wind surface and is maintained by the falling PW forcing that induces poleward/downward motion (and adiabatic warming over the pole).  The presence of the westward wind pocket over the polar region then markedly changes the wave forcing and circulation from climatology. First, the altered wind filters out much of the previously dominant westward GWD and allows eastward GWD to appear in the upper mesosphere. The missing westward GWD (and the related circulation) then causes the stratopause to separate downward from its climatological position. The presence of the anomalous eastward GWD reverses the climatological westward wind in the upper mesosphere to the eastward direction (a change that is out of phase with anomalies below). This forcing also induces strong upward motion that cools the lower mesosphere near the region where the stratopause previously existed."

**In article-** Temperature changes in the mesosphere and stratosphere connected with circulation changes in winter, Labitzke, 1972, on page 765: " **The stratopause descents 20 km** within several days **while the temperature around it rises rapidly** to a peak above 30 $^0$ C."

Finally, it is important to note that there is not a discontinuity in temperature across the tropopause or stratopause. Your analysis seems to require a large discontinuity across the boundary, but you are using layers 10s of km apart.

**Answer 6:** I have changed the term boundary. Instead, there is a term temperature discontinuity between the upper troposphere/lower stratosphere and upper stratosphere/lower mesosphere. I think this is closer to the real atmosphere and it is consistent with Marmolino, C., Severino, G., Deubner, F. L., and Fleck, B.: Phases and Amplitudes of Acoustic-Gravity Waves II. The Effects of Reflection, Astron. Astrophys., 278, 617-626, 1993.

Minor comments:

Line 14: "Temperature increases with ozone concentration". Ozone actually peaks in concentration in the lower stratosphere (e.g., Gotz, 1933), but the warming peaks at the top of the stratosphere due to the strong absorption of UV (e.g., Pendorf, 1936).

I changed this sentence in: "The temperature rises because solar energy is converted into kinetic energy when ozone molecules absorb ultraviolet (UV) radiation, leading to a warming of the stratosphere."

Line 20: Also consider citing Matsuno (1971)

I have included this reference.

Lines 21-22: "rapid descent and warming of the air in polar latitudes, mirrored by ascent and cooling above the warming." What altitudes?

Siskind, D. E., S. D. Eckermann, J. P. McCormack, L. Coy, K. W. Hoppel, and N. L. Baker (2010), Case studiesof the mesospheric response to recent minor, major, and extended stratospheric warmings, J. Geophys. Res., 115, D00N03,doi:10.1029/2010JD014114, on page 14: „ This removal of OGWD forcing from the stratosphere and mesosphere serves to remove a source of wave-driven mean down-welling **leading to cooling at 50 km.** "

Line 22: "…mirrored by ascent and cooling above the warming." Ref: Limpasuvan et al. (2016)

I have included this reference.

Line 24: "About six times per decade" Ref: Charleton and Polvani (2007)

I have included this reference.

Line 33: "These effects span both hemispheres" How so? You could point to changes in the summer mesospheric winds and gravity wave filtering (e.g., Gumbel and Karlsson 2011; Karlsson and Becker 2016; Körnich & Becker, 2010), inertial instability and growth of the summer hemisphere 2-day wave (Lieberman et al., 2023; France et al., 2018; Sato et al., 2023), and resulting polar mesopause warming and reduction in polar mesospheric clouds.

In article- Körnich, H., and E. Becker (2010), A simple model for the interhemispheric coupling of the middle atmosphere circulation, *Adv. Space Res., 45,* 661–68, https://doi.org/10.1016/j.asr.2009.11.001., page 4: "The GW drag is balanced by a residual circulation which extends as one global cell from the summer pole towards the winter pole. Lifting in the summer mesosphere and sinking in the winter mesosphere induces a temperature field that deviates strongly from the radiatively determined state. As a consequence, extremely cold temperatures at the summer mesopause give rise to the so-called noc-tilucent or polar mesospheric clouds."

Line 46: What did Cullens and Thurairajah (2021) find?

Their results indicate that although the main driver of SSWs are planetary waves, GWs can contribute to the occurrences and strength of SSWs.

Line 51-53: The two "important points" here aren't novel. It has been long understood that GWs play a critical role in the evolution of SSWs, e.g. Holton (1983), Liu (2017). Instead note how your findings provide new insight into how the temperature anomalies associated with SSW conditions modify the spectrum of GWs that propagate across the tropopause and stratopause.

I didn't say these points were new. They are listed in the Introduction, and the reader will find their

relevance in the Results and Discussion sections.

Lines 225-228: Is this relevant for the dynamically-driven, polar winter stratosphere, since it's in the dark?

In article- A climatology of stratopause temperature and height in the polar vortex and anticyclones, in JOURNAL OF GEOPHYSICAL RESEARCH, VOL. 117, D06116, doi:10.1029/2011JD016893, 2012 by France et al. on page 1: " Different physical processes maintain the stratopause at different latitudes and seasons. At sunlit latitudes, the stratopause is characterized by a temperature maximum near 50 km due to the absorption of shortwave radiation by ozone. In the polar night there is no solar insolation and a "separated" polar winter stratopause is maintained by gravity wave (GW) driven diabatic descent at high latitudes [e.g., Hitchman et al., 1989]."

Minor Changes:

Line 25: change "devided" to "divided"

I have changed this.

Line 103-104: Consider rewording this sentence.

I have changed this sentence in: "Dimensionless equations are used because of their applicability to various stratified media, including the Earth's atmosphere, planetary atmospheres, and the solar atmosphere."

Brunt–Väisälä is misspelled in the Figure 1 caption and on Line 86.

I have changed this.

Line 190: "atmosphere" is misspelled

I have changed this.

**P. S. The revised manuscript in PDF is attached as supplementary material.**

---

## Author Comment (AC2)

This paper studies the coupling between the troposphere, stratosphere, and mesosphere by gravity waves during a sudden stratospheric warming (SSW). By using an analytical reflection coefficient R to characterise the spectrum of waves that is transmitted/reflected at the tropopause and stratopause during SSW events, the study offers a novel perspective on the role of gravity waves in atmospheric coupling and the observed effect of a cooling mesosphere during an SSW. Given the knowledge gaps in this area and the need for better representation of gravity wave effects in models, I believe this paper adds value and recommend it for publication provided that the following comments are addressed.

- Whilst the data in the figures supports the conclusion drawn regarding the changes in R during this idealised SSW, I am confused by the key facts used to illustrate the takeaway points. Firstly, please could you provide a reason for the R<0.4 threshold used to delineate transmitted/reflected waves in figures 2 – 5, it seems like an arbitrary choice. Please then could you provide a reason for the $\Omega$ = 0.2 and $\Omega$ = 0.1 thresholds in Figs 2/4/5 and Fig 3, respectively? $\Omega$ < 0.2 and $\Omega$ < 0.1 mark most likely transmitted waves given R = 0.4, however the $\Omega$ thresholds don't appear to reflect what the figure tells you. For example, in Fig. 2, almost all $\Omega$ have R < 0.4 for Vh < 0.5, the text in line 149 however implies that only $\Omega$ < 0.2 have R < 0.4 for Vh < 0.9. The same critique applies to Fig 3 and L 155, Fig 4 and L 170, Fig 5 and L 177. These numbers are used in the discussion in the paper, so are a significant aspect which needs addressing before the paper is published.

These thresholds really have no physical meaning. When I drew these Figs. I thought it might be easier for comparison between Figs. to mark some frequencies and reflection coefficient values. Therefore, I chose R=0.4 and $\Omega$ < 0.2. In this version of the article, I have deleted these thresholds.

- The assumptions used in deriving the reflection coefficient, and the set-up of the two scenarios: no-SSW, and SSW, are highly idealised. I.e., The atmosphere from the tropopause to the mesosphere is not isothermal and is highly variable compared to the two idealised scenarios given. Please could you address the sensitivity of your results to deviations from the assumptions and idealisations, and the validity of using this form of the reflection coefficient to the real atmosphere? Perhaps you could support its validity by calculating R from data of observed or modelled SSW events, which should hopefully show that R increases at the tropopause, decreases at the stratopause, and in such a way that suggest similar changes in gravity wave fluxes to those reported in this paper.

In the article the atmosphere from the tropopause to the mesosphere is not isothermal because there is a temperature stratification from the upper troposphere/lower stratosphere with T=220 K to the middle stratosphere with T=240 K. The similar temperature stratification is between the middle stratosphere (T=240 K) and upper stratosphere/lower mesosphere with T=270 K, this is no-SSW case. During the SSW, there are temperature changes in the stratosphere and mesosphere and temerature stratification is different:  the upper troposphere/lower stratosphere has the same temperature T=220 K, while the temperature in the middle atmosphere increases for 25 K, T=265 K. The temperature of the upper stratosphere/lower mesosphere also changed. This part of the atmosphere cools and the temperature decreases for 25 K, to T=245 K. This is the same approach as in the article -Phases and amplitudes of acoustic-gravity waves II: The effects of reflection, Astronomy and Astrophysics. 278, 617-626 (1993) by Marmolino el al. in abstract: " **We study wave reflection caused by the temperature stratification of the solar atmosphere, assumed to be a succession of two layers of different temperatures...**" Page 620, Fig. 2: "Amplitudes and phases of the reflection and transmission coefficients R and T vs. frequency at $k_x$=1.31 Mm$^{-1}$. Solid lines refer to **a succession of two layers suitable to represent the photospheric stratification ($T_l$=5000 K** (this means T in the lower layer); $T_u$=**4500 K** (this means T in the upper layer)..."   The same is in my article

where the succession of the two layers with temperatures $T_1$=220 K and $T_2$=240 K represents the atmospheric stratification between the upper troposphere/lower stratosphere and middle stratosphere; also the temperatures $T_2$=240 K and $T_3$=270 K represent the stratification between the middle stratosphere and the upper stratosphere/lower mesosphere.

I can support my results because in these articles the results are the same as mine: Namely, my Fig. 3 shows that SSW events prevent GWs propagation from the troposphere towards the stratosphere, which is consistent with known scientific results:

Wang, L. and Alexander, M. J.: Gravity wave activity during stratospheric sudden warmings in the 2007–2008 Northern Hemisphere winter, J. Geophys. Res.-Atmos., 114, D18108, https://doi.org/10.1029/2009JD011867, 2009;

Hindley, N., Wright, C., Hoffmann, L., Moffat-Griffin, T., and Mitchell, N.: An 18 year climatology of directional stratospheric gravity wave momentum flux from 3-D satellite observations, Geophys. Res. Lett., 47, https://doi.org/10.1029/2020GL089557, e2020GL089557, 2020;

Wicker, W., Polichtchouk, I., and Domeisen, D. I. V.: Increased vertical resolution in the stratosphere reveals role of gravity waves after sudden stratospheric warmings, Weather Clim. Dynam., 4, 81–93, https://doi.org/10.5194/wcd-4-81-2023, 2023.

Also, the absence of gravity wave breaking in the mesosphere explains the mesospheric cooling during an SSW as in:

Holton, J. R.: The influence of gravity wave breaking on the general circulation of the middle atmosphere, Journal of the Atmospheric Sciences, 40, 10, doi:10.1175/1520-0469(1983)040<2497:TIOGWB>2.0.CO;2, 1983;

Liu, H. L. and Roble, R. G.: A study of a self-generated stratospheric sudden warming and its mesospheric-lower thermospheric impacts using the coupled TIME-GCM/CCM3, J. Geophys. Res. 107, 18, doi:10.1029/2001JD001533, 2002.

Stephan, C. C., Schmidt, H., Zuelicke, C., and Matthias, V.: Oblique gravity wave propagation during sudden stratospheric warmings, Journal of Geophysical Research: Atmospheres, 125, e2019JD031528, https://doi.org/10.1029/2019JD031528, 2020; citation: "During the course of the SSW the mesospheric GW momentum flux (GWMF) turns from mainly westward to mainly eastward. **Waves of large phase speed (40–80 m s-1 ) dominate the eastward GWMF** during the peak phase of the warming. In addition, the contribution of slow phase speed waves to the total GWMF decreases dramatically and waves of phase speeds > 40 m s-1 dominate the eastward GWMF during the peak phase of the warming. My result for the phase velocity of gravity waves that propagate from the stratosphere to the mesosphere during SSW is 65 ms-1<Vh<280 ms-1. These waves have the best chance to pass the upper stratosphere/lower mesosphere temperature discontinuity.

**In article**- Dörnbrack, A., Gisinger, S., Kaifler, N., Portele, T. C., Bramberger, M., Rapp, M., Gerding, M., Faber, J., Žagar, N., and Jelić, D.: Gravity waves excited during a minor sudden stratospheric warming, Atmos. Chem. Phys., 18, 12915–12931, https://doi.org/10.5194/acp-18-12915-2018, 2018, page 12921: "Applying this relationship results in an estimate of the scaled intrinsic frequency being /f ≈ 1.4–1.7 for the stratospheric layers on 30 January. This means that the observed waves are dominated by intrinsic frequencies much smaller than the **buoyancy frequency N ∼= 0.02 s −1**." My result is that gravity waves with frequency of ω < 0.005s −1 have the best chance of propagating from the troposphere. This frequency is 4 times smaller than N ∼= 0.02 s −1.

**In article**- High Resolution Dynamics Limb Sounder observations of the gravity wave-driven elevated stratopause in 2006, by France, J. A. Et al., JOURNAL OF GEOPHYSICAL RESEARCH, VOL. 117, D20108, doi:10.1029/2012JD017958, 2012, page 7: " There is **an increase in GW KMF** beginning on 5 January that maximizes on 8 January, **which extends from the lower stratosphere to the lower mesosphere.** The largest amplitudes of GW KMF (gravity wave kinetic momentum flux) on this date occur in the lower mesosphere. The temperature contours indicate an increase in temperature at the stratopause following the increase in GW KMF between 5 and 8 January." My Fig. 5 shows that during SSW gravity waves have lower reflection coefficient values and a higher chance of propagating from the stratosphere to the mesosphere.

**The same is in article-** Satellite observations of middle atmosphere gravity wave absolute momentum flux and of its vertical gradient during recent stratospheric warmings, Atmos. Chem. Phys., 16, 9983–10019, https://doi.org/10.5194/acp-16-9983-2016, 2016, by Ern, M., et al, on page 10000: " Finally, it should be mentioned that, in all winters considered, enhanced values of gravity wave potential drag are preferentially found in the upper stratosphere and in the mesosphere (i.e., at altitudes above about 40 km)."

The waves with frequencies in my article are found in the atmosphere at the altitudes about 55-60 km, as in article Nina, A. and Čadež, V.: Detection of acoustic-gravity waves in lower ionosphere by VLF radio waves, Goephysical Research Letters, 40, 18, 4803-4807, https://doi.org/10.1002/grl.50931, 2013. These waves are presented in Fig. 3, bottom, on page 4805. These waves are gravity waves with frequencies lower than Brunt–Väisälä frequency of $0.02$ s$^{-1}$. In my article gravity waves that propagate from the stratosphere to the mesosphere have a frequencies lower than $0.019$s$^{-1}$.

About the deviations from the assumptions and idealizations- It is known that -..." at the bottom side of the thermal layers (where the temperature profile is increasing in altitude) the positive temperature gradient with increasing altitude signifies an increase in atmospheric stability and reduction of vertical mixing. In contrast, at the topside of the thermal layer the negative temperature gradient with decreasing altitude implies a reduction in atmospheric stability to the point that the atmosphere may become convectively unstable, thus possibly supporting the development of turbulence," page 4 in MESOSPHERE INVERSION LAYERS AND STRATOSPHERE TEMPERATURE ENHANCEMENTS by John W. Meriwether and Andrew J. Gerrard, Reviews of Geophysics, 42, RG3003 / 2004. Namely, the turbulence can not be described by the linear theory applied in my article. In addition, the potential for the nonlinear interaction also exists. This can induce gravity wave breaking at lower altitudes in the atmosphere and also the generation of the secondary gravity waves ( Gavrilov, N. M. and Kshevetskii, S. P.: Identification of spectrum of secondary acoustic-gravity waves in the middle and upper atmosphere in a high-resolution numerical model, Solar-Terrestrial Physics, 9, 3, 86-92, doi: 10.12737/stp-93202310, 2023.). Linear approximation can be understood as a limitation of this work.

- It is not clear how equations 5 come from equations 1 – 3, this appears to be a novel step in deriving the acoustic gravity wave dispersion relation, so would be insightful to have the steps elaborated on with their physical meaning. The reader is pointed to more detail in Jovanovic (2016), however no extra details to what is shown in this paper are given.

This is not a novel step. This procedure is used in Pinter, B. Čadež, V. M., and Roberts, B: Waves and instabilities in a stratified isothermal atmosphere with constant Alfven speed – revisited, Astron. Astrophys. 346, 190–198, 1999. I used the same approach in the Sect. 2. Also in the article – Diagnostics of plasma in the ionospheric D-region: detection and study of different ionospheric

disturbance types, A. Nina, V. M. Čadež , L.  Č. Popović, and V. A. Srećković, in The European Physical Journal, Vol. 71, Issue 7, 2017.

According to your comments I include some new details in this section.

- Please ensure that all symbols are correctly defined. E.g., no variables in equations (1 – 3) are defined except the constants.

I did it. I thought it wasn't necessary because p, T, and ρ have usual meanings.

- For brevity, it would probably be more appropriate to refer to the troposphere-stratosphere boundary as the tropopause and the stratosphere-mesosphere boundary as the stratopause. All terms appear in the paper, so I suggest picking a single convention for consistency in the paper.

The term boundary is important because of the model in Sect. 3. On the other hand, in the real atmosphere there is a tropopause and a stratopause between the troposphere/stratosphere and between the stratosphere/mesosphere, respectively. I have changed the boundary term in Section 4 (Results) to include atmospheric temperature stratification. Thererfore, there is no boundary between the troposphere and stratosphere (nor tropopause); there is only a temperature discontinuity between the upper troposphere/lower stratosphere with a temperature of 220 K and middle  stratosphere with a temperature of 240 K. The same is for the stratosphere/mesosphere-there is no stratopause, just a temperature discontinuity between the  middle  stratosphere with a temperature of 240 K  and the upper stratosphere/lower mesosphere with a temperature of 270 K in case without SSW. As suggested by another reviewer, I need to include the term stratopause in the discussion because during SSW events the stratopause drops from about 55 km to about 40 km and heats the middle and lower stratosphere.

- Please could the labels on the figures be made in a larger font, at least to match the font size in the paper.

The Figs. 2, 3, 4, 5 are made in mini page format with a width of 0.48 cm. This allows the Figs. to be placed side by side, which is convenient for comparing them. I changed their width to 0.494 cm. Its a maximum  width for the minipage Figs. Of course, it is always possible to make larger Figs., but they won't be side by side.

- Please could you also review the spelling and grammar, as many instances of grammatical errors were spotted in the paper.

I tried to do that.